# The assembly of the Mitochondrial Complex I Assembly complex uncovers a redox pathway coordination

Lindsay McGregor [1], Samira Acajjaoui[1], Ambroise Desfosses [2], Melissa Saïdi[1], Maria Bacia-Verloop[2], Jennifer J. Schwarz[3], Pauline Juyoux[4], Jill von Velsen [4], Matthew W. Bowler[4], Andrew A. McCarthy [4], Eaazhisai Kandiah [1], Irina Gutsche [2,5] ✉ & Montserrat Soler-Lopez [1] ✉

The Mitochondrial Complex I Assembly (MCIA) complex is essential for the biogenesis of respiratory Complex I (CI), the first enzyme in the respiratory chain, which has been linked to Alzheimer's disease (AD) pathogenesis. However, how MCIA facilitates CI assembly, and how it is linked with AD pathogenesis, is poorly understood. Here we report the structural basis of the complex formation between the MCIA subunits ECSIT and ACAD9. ECSIT binding induces a major conformational change in the FAD-binding loop of ACAD9, releasing the FAD cofactor and converting ACAD9 from a fatty acid β-oxidation (FAO) enzyme to a CI assembly factor. We provide evidence that ECSIT phosphorylation downregulates its association with ACAD9 and is reduced in neuronal cells upon exposure to amyloid-β (Aβ) oligomers. These findings advance our understanding of the MCIA complex assembly and suggest a possible role for ECSIT in the reprogramming of bioenergetic pathways linked to Aβ toxicity, a hallmark of AD.

A worldwide research effort is currently underway to identify the factors driving Alzheimer's disease (AD) pathogenesis in order to find better ways to diagnose the disease, delay its onset and prevent progression. Due to their high energy demands and limited glycolysis rate, neurons depend on an efficient oxidative metabolism—i.e., oxidative phosphorylation (OXPHOS) and fatty acid beta-oxidation (FAO)—and are particularly vulnerable to mitochondrial dysfunctions. Because of this, age-related degradation of mitochondria is a prime suspect in AD pathophysiology[1,2]. Notably, in the brain of AD patients amyloid-β (Aβ) peptides progressively accumulate within mitochondria and perturb the mitochondrial respiratory Complex I (CI)[3], a ~1 MDa membrane protein complex composed of 44 different subunits (encoded in the mitochondrial and nuclear genome) that is essential for OXPHOS[4]. Despite its central importance, how CI is altered by Aβ and relates to neuronal integrity remains an open question.

While the molecular structure of the core CI subunits has been determined in atomic detail[5], much less is known about the biogenesis of CI. This multistep process involves transiently associated assembly factors that integrate core and accessory/supernumerary subunits as well as cofactors into the final holoenzyme. A key player in CI biogenesis is the mitochondrial CI assembly (MCIA) complex[6]. This complex consists of three core proteins—NDUFAF1, ACAD9 and ECSIT—that appear to further associate with three peripheral membrane proteins[6]. The organisation of the MCIA complex and its role in CI assembly are unclear, partly because the individual MCIA components also mediate other cell functions. In particular, ACAD9 was annotated as an acyl-CoA dehydrogenase (ACAD) enzyme due to its sequence

[1]Structural Biology Group, European Synchrotron Radiation Facility (ESRF), 38043 Grenoble, France. [2]Institut de Biologie Structurale, Université Grenoble Alpes, CEA, CNRS (IBS), 38044 Grenoble, France. [3]European Molecular Biology Laboratory (EMBL), 69117 Heidelberg, Germany. [4]European Molecular Biology Laboratory (EMBL), 38043 Grenoble, France. [5]Department of Chemistry, Umeå University, Umeå, Sweden. ✉e-mail: irina.gutsche@ibs.fr; montserrat.soler-lopez@esrf.fr

homology (47% amino acid identity and 65% amino acid similarity) with the very long chain acyl-CoA dehydrogenase VLCAD[7], which both can initiate the FAO pathway with the concurrent reduction of their FAD cofactor[8]. In turn, ECSIT participates in cytoplasmic and nuclear signalling pathways, in which different ECSIT isoforms and post-translational modifications might contribute to its functional complexity[9]. Furthermore, ECSIT was identified as a molecular node interacting with Aβ-producing enzymes[10], potentially implicating it in AD pathogenesis[11].

We and others recently discovered that C-terminal ECSIT binding triggers ACAD9 deflavination, switching it from an FAO enzyme to a CI assembly factor[12,13]. These two mutually exclusive functions allow for the coordinated regulation between distinct multifunctional protein complexes to ensure efficient energy production. Here we provide high resolution structural insights into the MCIA subcomplex, ACAD9$_{WT}$-ECSIT$_{CTER}$, and identify the conformational changes that ACAD9 undergoes upon ECSIT binding. We provide experimental evidence of in vitro and ex cellulo ECSIT phosphorylation and its effect on the ACAD9-ECSIT interaction and reveal a decrease in ECSIT phosphorylation levels upon exposure to Aβ oligomer toxicity. These findings will pave the way for evaluating MCIA proteins as potential diagnostic biomarkers for detecting early AD pre-symptomatic stages when mitochondria are primarily affected by Aβ toxicity[14].

## Results

### A 15-residue ECSIT peptide is essential for complex formation with ACAD9

We previously reported a cryo-electron microscopy (cryo-EM) structure of ACAD9 in complex with a C-terminal fragment of ECSIT (ECSIT$_{CTD}$, residues 247–431) at low (-15 Å) resolution[12]. The inclusion of an additional N-terminal 26 residues in the ECSIT construct (hereafter named ECSIT$_{CTER}$, residues 221–431), combined with optimisation of the purification and subcomplex reconstitution protocols (Supplementary Fig. 1A–D) led to a 3.0 Å resolution cryo-EM structure of the ACAD9$_{WT}$-ECSIT$_{CTER}$ complex (Fig. 1, Supplementary Figs. 3, 4A, 5A–D and Supplementary Table 1), enabling us to build and refine an atomic model that includes nearly all ACAD9 residues as well as a 15-residue peptide spanning ECSIT residues 320–334 (Fig. 1A, B, E). As previously described[15], ACAD9 is present as a dimer, where the C-terminal regions (residues 600–621) form helix loops that are positioned around the neighbouring protomer thereby stabilising the active dimer (Fig. 1B, C). Such helix swapping has been suggested to promote the stability of the homodimer in structural homologues adopting the same C2-symmetry[16]. The ACAD9 monomer consists of an N-terminal dehydrogenase domain (res. 38–453 comprising an α-helical, a β-sheet and a second α-helical subdomains) and a C-terminal α-helical bundle vestigial domain (res. 487–587) connected by a poorly ordered -35-residue linker (Fig. 1A, C). As expected, no significant density was observed for the flavin adenine dinucleotide (FAD) in the dehydrogenase domain, confirming our previous observation that the binding of ECSIT triggers the deflavination of ACAD9[12].

The ECSIT binding site is located at the junction of the ACAD9 dehydrogenase and vestigial domains (Fig. 1A–D), with one ECSIT$_{CTER}$ per ACAD9 protomer. ECSIT residues 322–328 form a 3$_{10}$-helix that interacts with the ACAD9 β1–β2 loop, while ECSIT residues 329–334 adopt an extended loop conformation lying in anti-parallel orientation between the ACAD9 α16 and η6 helices (Fig. 1C, D). Examination of the electrostatics properties of the binding interface shows that the surface of the ACAD9 is mainly positively charged whereas the modelled ECSIT sequence is mostly negative (Fig. 1F). Details of ECSIT$_{CTER}$ recognition by the ACAD9 dehydrogenase/vestigial interface, showing key binding interactions, are summarised in Fig. 1D.

Mutations within the ECSIT helix exhibited distinct characteristics upon analysis of the ACAD9$_{WT}$-ECSIT$_{CTER}$ complex formation, emphasising the specificity of the ACAD9-ECSIT interaction. In

particular, an ECSIT-E323A mutation underscores the significance of the salt bridge formation between ECSIT Glu323 and ACAD9 Lys228, resulting in the absence of binding, as evidenced by mass photometry and DLS (Fig. 2A, H). The relevance of the pyrrole-containing aromatic sidechain and hydrogen bond between ECSIT Trp324 and ACAD9 Ser191, located on the tip of the β1–β2 loop, was demonstrated by a reduction in complex formation of ACAD9 with an ECSIT-W324A mutant (Fig. 2B, H). An ACAD9-S191A mutant further confirmed the lower affinity for ECSIT$_{CTER}$ (Supplementary Fig. 2A and Fig. 2G–I), though this mutant showed higher stability compared to ACAD9$_{WT}$, with less dissociation into monomers (Supplementary Fig. 1B and Fig. 2G). Interestingly, despite their positioning in the ACAD9 binding pocket (Fig. 1D, E), ECSIT residues Tyr327 and Tyr328 do not appear to strongly interact with any ACAD9 residues (Fig. 1D). Replacement of these Tyrosine residues by either Phenylalanines or Alanines shows that the loss of the hydrophobic nature/π-bonding and hydrogen bonding capabilities of Tyr327 has a more pronounced impact on complex formation than the removal of the hydrogen bonding partner only, whereas the removal of both the hydrophobic/π-bonding and hydrogen bonding properties of Tyr328 permitted complex formation (Fig. 2C–F, H). In summary, it can be deduced that while all interactions play a significant role, the stability of the complex primarily relies on the salt bridge formation between ECSIT residue Glu323 and Lys228 of ACAD9, as well as the crucial hydrogen bonds established between Trp324 of ECSIT and ACAD9 residues Ser191/Asp188 (Supplementary Table 2).

For additional insights, we used AlphaFold2 (AF2)[17] to predict the structure of the ACAD9-ECSIT complex using sequences matching our constructs. The AF2 model predicted the binding site to contain ECSIT residues 316–338 and closely resembled the experimental structure (RMSD of 1.635 Å over 552 Cα atoms; Supplementary Fig. 6A–C). No additional ECSIT residues were predicted to contact ACAD9, suggesting that most, if not all, of the specific interactions between these proteins were captured in the cryo-EM structure. Indeed, we found that a synthetic peptide spanning ECSIT residues 318–336 was sufficient to cause the deflavination of ACAD9 (Supplementary Fig. 2B), confirming that these residues are crucial for complex formation. Notably, the full length AF2 model available from the database predicts that ECSIT comprises two flexibly linked globular domains, an N-terminal domain (res. 70–208) with a pentatricopeptide repeat fold and a C-terminal domain (res. 221–392) with an RNaseH-like fold comprising a six-stranded mixed β-sheet and four short helices[18]. The ACAD9-binding residues of ECSIT reside within the C-terminal domain and localise to the tip of a long loop (res. 306–357) containing a 3$_{10}$-helix that is flexibly connected to strands β4 and β5 of the RNaseH-like domain. This turns to be very close to a density extension observed in our earlier low-resolution map of ACAD9-ECSIT$_{CTD}$[12] (Supplementary Fig. 6D, E), allowing this domain to be positioned within the extra density with only a minor adjustment of the long predicted β4-β5 loop. The long linkers (residues 306–320 and 335–356) (Supplementary Fig. 6A, D) connecting the ACAD9 binding region of ECSIT to the RNaseH-like C-terminal domain likely contribute to the high flexibility observed in biophysical analyses[12] (Fig. 1A and Supplementary Fig. 6D).

### ECSIT opens the FAD-cofactor gatekeeper loop of ACAD9 upon binding

In order to identify structural changes that occur in ACAD9 upon ECSIT binding, we pursued a cryo-EM structural study of ACAD9$_{WT}$ in the unbound state. Given a strongly preferred orientation (Supplementary Fig. 5A, B) resulting in an anisotropic 3D reconstruction and limiting the resolution of the final map to ~5.0–5.5 Å in the core and ~6.5–8.0 Å in the periphery (Supplementary Fig. 5E, F), the analysis was restricted to a rigid body fit of the ACAD9 AF2 model (residues 38–621) (Supplementary Fig. 4B). Considering the observed stability and compactness of the ACAD9$_{S191A}$ mutant (Fig. 2G, H), we chose it as a

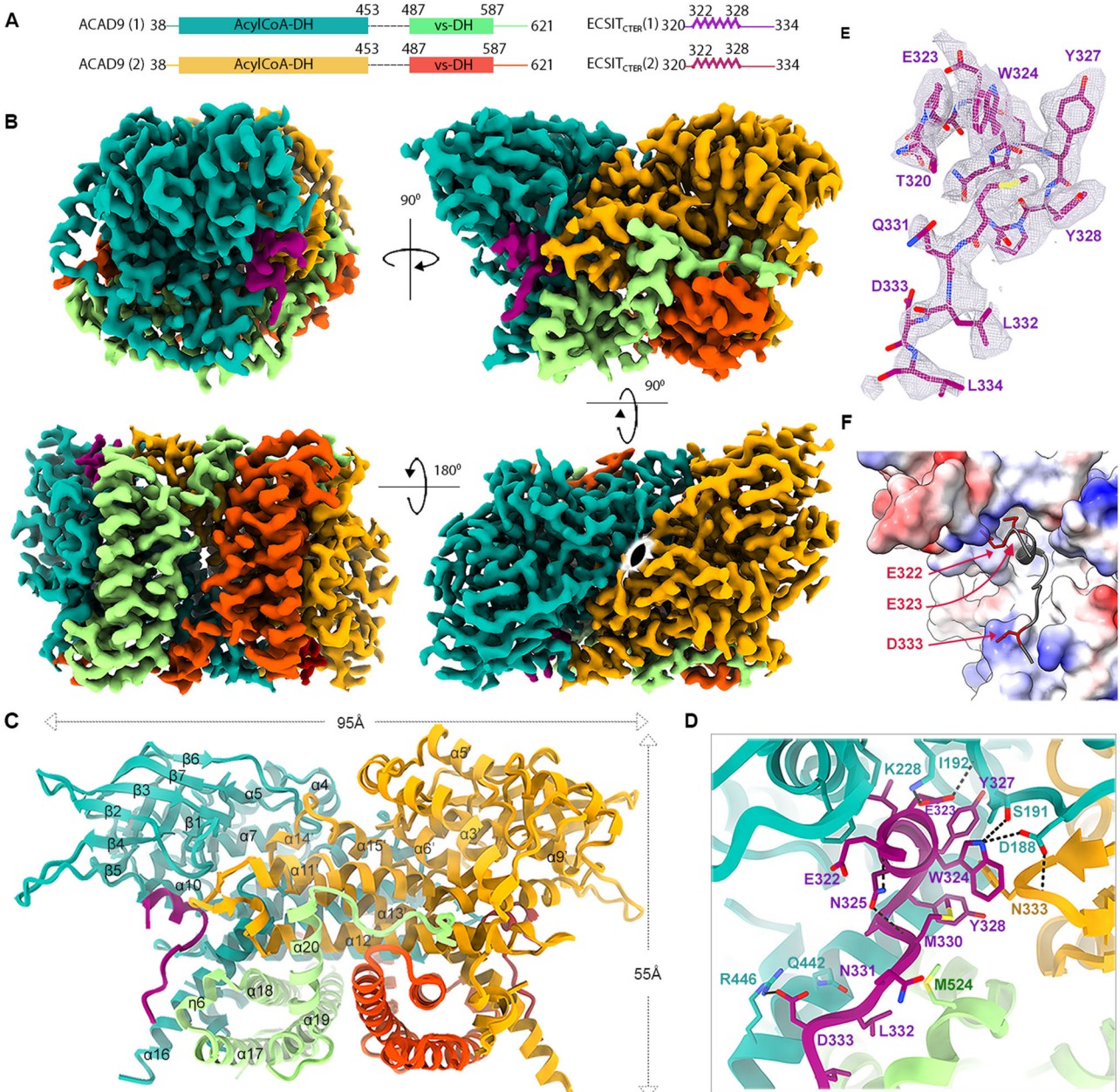

**Fig. 1 | Cryo-EM structure of the ACAD9$_{WT}$-ECSIT$_{CTER}$ complex. A** Schematic of the ACAD9 and ECSIT$_{CTER}$ sequences modelled in the cryo-EM map. The two ACAD9 monomers are labelled (1) and (2), with their dehydrogenase domains shown in teal and light orange, and their vestigial domains in light green and dark orange, respectively. ECSIT$_{CTER}$ monomers are in purple and red. The dashed lines indicate ACAD9 residues 454–486 absent from the final model. **B** Cryo-EM map of ACAD9$_{WT}$-ECSIT$_{CTER}$ showing side, front, top and bottom views (clockwise from top left) coloured domain-wise as in (**A**), with C2 symmetry represented (bottom right). **C** Ribbon diagram of the refined structure of the ACAD9$_{WT}$-ECSIT$_{CTER}$ complex coloured as in (**A**). **D** Key interactions involved in ECSIT$_{CTER}$ recognition by the ACAD9 dehydrogenase/vestigial interface. Specifically, ECSIT Glu323 makes a salt bridge with ACAD9 Lys228, stabilising the ECSIT binding on the N-terminal segment with the ACAD9 β4 strand, whereas a second salt bridge between ECSIT Asp333 and ACAD9 Arg446 stabilises the ECSIT binding on the C-terminal end with the ACAD9 α16 helix. Furthermore, ECSIT directly interacts with the ACAD9 β1–β2

loop through three hydrogen bonds between ECSIT Trp324 and ACAD9 Ser191 and Asp188, respectively, and ECSIT Glu323 with the backbone amide of Ile192. Notably, while there is no evidence of direct ECSIT interactions with the second ACAD9 protomer, Asp188 shows an additional hydrogen bond with Asn333 on α11′–α12′ loop, bridging ECSIT binding with the ACAD9 dimer. Numerous intramolecular hydrogen bonds within ECSIT contribute to the stability of the conformation. In particular, Thr320 forms a hydrogen bond with Gln331 while Asn325 interacts with the backbone amide of Glu322 and Met330, respectively. **E** Close-up of the cryo-EM map of the ECSIT$_{CTER}$ peptide. **F** Representation of the electrostatic properties at the ACAD9$_{WT}$-ECSIT$_{CTER}$ binding interface as calculated by APBS electrostatics. The surface of the junction between the vestigial and dehydrogenase domains of ACAD9 is mainly positively charged (blue). In comparison, the ECSIT sequence modelled from the cryo-EM map contains hydrophobic and polar residues (grey) but principally carries a negative charge. Negatively charged residues Glu322, Glu323 and Asp333 are represented as red sticks for clarity.

suitable candidate for further improving the ACAD9 structure. To increase the orientational diversity, we collected a 35° tilted cryo-EM data set (Supplementary Fig. 5A, B) and solved the structure of ACAD9$_{S191A}$ at 3.6 Å resolution (Supplementary Figs. 4C, 5G, H, 7A–C). Consistent with FAO activity, the structure features one molecule of

FAD in the cofactor binding site of each ACAD9 protomer (Supplementary Fig. 7C, D). As expected, no significant density occupies the ECSIT binding site identified in the ACAD9$_{WT}$-ECSIT$_{CTER}$ structure.

Strikingly, comparing the structures of ACAD9 alone and in complex with ECSIT$_{CTER}$ reveals that, in addition to causing

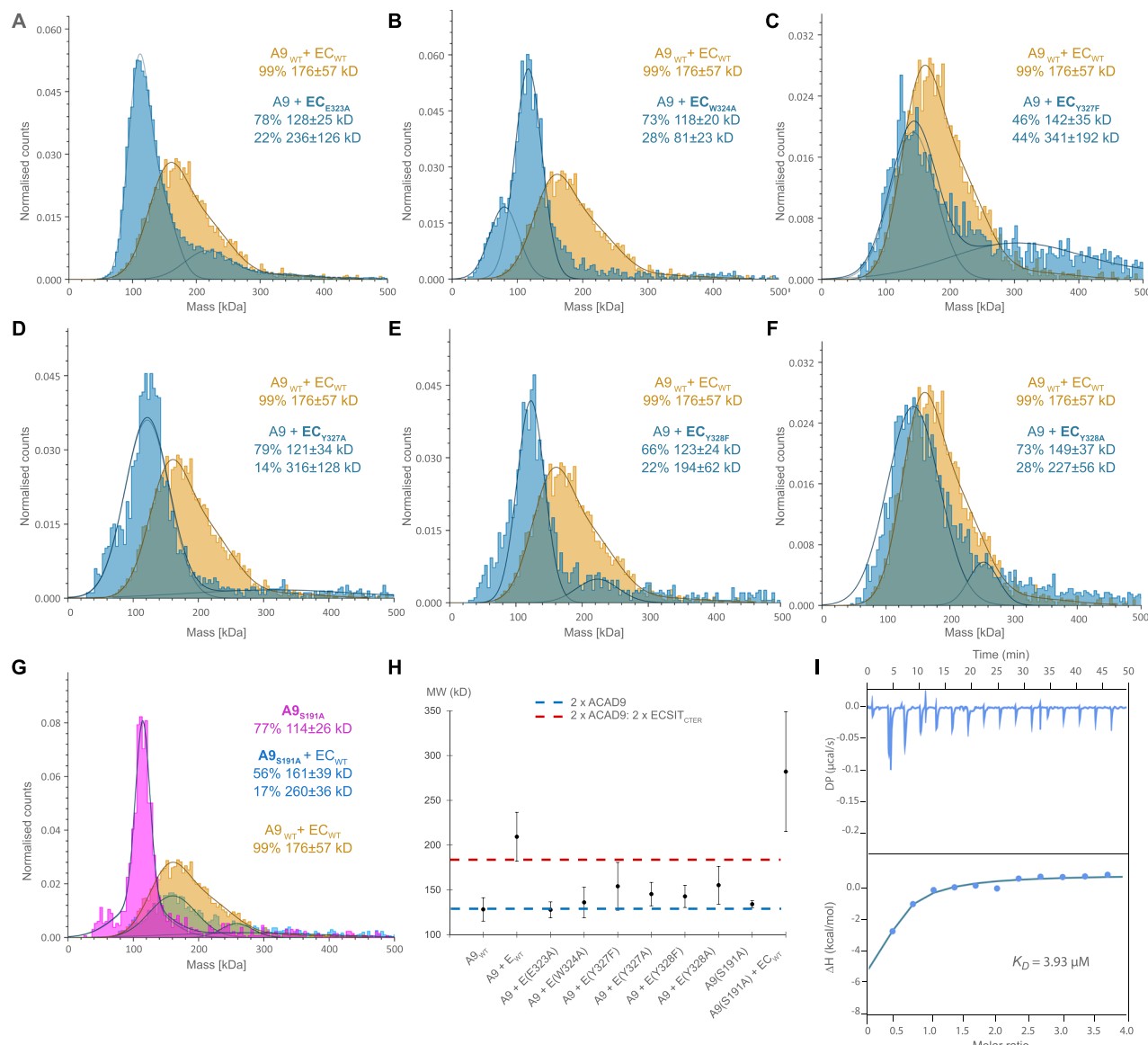

**Fig. 2 | Analysis of interface derived mutants on ACAD9$_{WT}$-ECSIT complex formation.** Several mutants were designed to probe the complex interface and identify the role of the residues in complex formation. The mutant species were reconstituted with the corresponding WT protein and compared with ACAD9$_{WT}$-ECSIT$_{CTER}$. **A** Mass photometry assays show that ECSIT$_{CTER-E323A}$ severely impacts the formation of the complex, with no MW corresponding to ACAD9-ECSIT$_{CTER-E323A}$ observed. **B** ECSIT$_{CTER-W324A}$ negatively impacts complex formation, with the MW of the main species similar to ACAD9$_{WT}$. **C** ECSIT$_{CTER-Y327F}$ complexes with ACAD9$_{WT}$ but also forms higher order species. **D** ECSIT$_{CTER-Y327A}$ affects complex formation similarly to (**B**). **E** ECSIT$_{CTER-Y328F}$ has a detrimental effect on complex formation as (**B**) and (**D**) whereas (**F**) ECSIT$_{CTER-Y328A}$forms the complex, however, with additional higher MW species. **G** ACAD9$_{S191A}$ appears a more stable dimer than ACAD9$_{WT}$ (Supplementary Fig. 1B) and forms an ACAD9$_{S191A}$-ECSIT$_{CTER}$ complex.

Experiments were repeated thrice with similar results. **H** DLS assays show that all ECSIT mutants show a reduced average MW (130–150 kDa) in comparison to the ACAD9$_{WT}$-ECSIT$_{CTER}$ complex, indicating partial complex formation but a reduction in stability. ACAD9$_{S191A}$ has a MW very similar to ACAD9$_{WT}$, however, in complex with ECSIT$_{CTER}$, there is the formation of higher MW species. Blue and red dashed lines indicate the expected MW for ACAD9$_{WT}$- homodimer and ACAD9$_{WT}$-ECSIT$_{CTER}$ complex, respectively. Data are presented as mean values ± SD of at least $n = 3$ biological replicas. **I** ITC binding assay for the binding affinity between ACAD9$_{S191A}$ and ECSIT$_{CTER}$. The equilibrium dissociation constant ($K_D$) of the ACAD9$_{S191A}$-ECSIT$_{CTER}$ complex is 3.93 μM, ~3-fold lower than that of ACAD9$_{WT}$-ECSIT$_{CTER}$ (Supplementary Fig. 2A). The experiment was repeated thrice with similar results. Source data are provided as a Source Data file.

deflavination, ECSIT binding induces a large conformational change in the β1–β2 loop adjacent to the FAD binding site (Figs. 3 and 5D, E). In particular, the structure of ACAD9$_{S191A}$ shows that this loop adopts a closed, downward facing position, forming hydrogen bonds with the α11-α12 loop of the neighbouring monomer (Supplementary Fig. 7C) and thereby acting as a barrier between the adenine nucleotide of the FAD molecule and the external solvent (Fig. 3A, C and Supplementary Fig. 8A, D). In contrast, the loop adopts a very different conformation in the ACAD9$_{WT}$-ECSIT$_{CTER}$ model (Fig. 3B, D,

E and Supplementary Fig. 8B, E), where the tip of the β1-β2 loop (residue Gly186) moves by ~10 Å upwards to allow the ECSIT 3$_{10}$-helix to insert, suggesting that this mobile ACAD9 β1-β2 loop plays the role of a gatekeeper (Fig. 3F). Although it does not directly interact with the FAD molecule, we suggest that removal of this barrier destabilises the cofactor environment, leading to deflavination and the reassignment of ACAD9 from an FAO enzyme to a CI assembly factor. The downward facing loop position in ACAD9$_{S191A}$ clashes with the superposed ECSIT 3$_{10}$-helix indicating that the flipping mechanism is

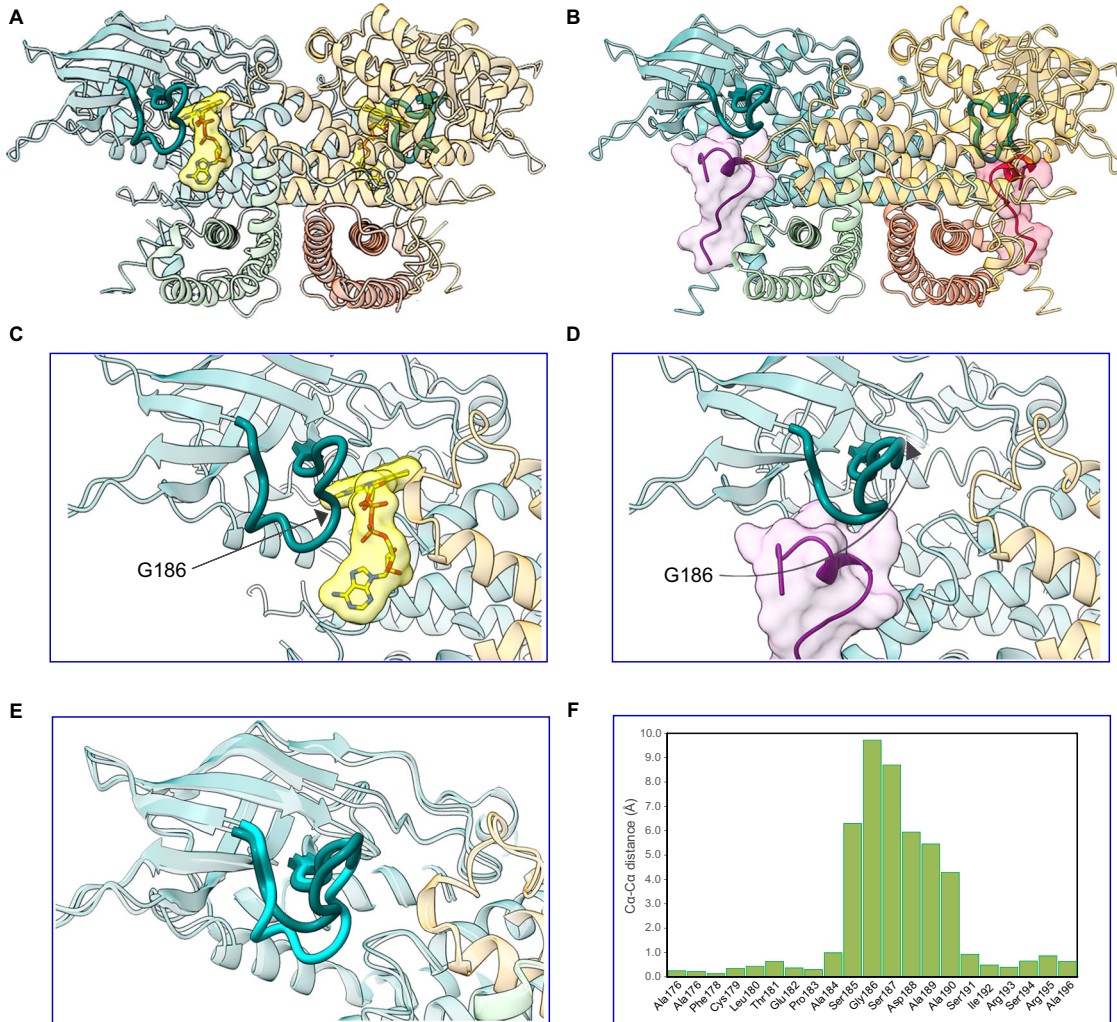

**Fig. 3 | A gatekeeper loop movement of 10 Å is induced upon ECSIT$_{CTER}$ binding to ACAD9.** ECSIT binding to ACAD9 induces the deflavination of ACAD9 and the displacement of a loop bridging the ECSIT:FAD binding sites. **A** ACAD9$_{S191A}$ in dehydrogenase form, with the bound FAD highlighted in yellow. The β1–β2 loop consisting of ACAD9 residues 178–195 is depicted in dark teal. **B** ACAD9$_{WT}$-ECSIT$_{CTER}$ complex showing the location of the two ECSIT binding sites adjacent to ACAD9 residues F178-R195 in bold. **C** Close-up of (**A**), showing the loop adopting a

downfacing position, in a closed conformation, acting as a barrier to the internal core of ACAD9 and the FAD pocket. **D** Close-up of (**B**), after ECSIT binds to ACAD9. Residue Gly186 is on the tip of the β1–β2 loop and is displaced by ~10 Å, flipping upwards into an open conformation. **E** Superposition of the β1–β2 loop in ACAD9 unbound (cyan) and in complex with ECSIT$_{CTER}$ (dark teal). **F** Cα-Cα distance positions of the β1–β2 loop residues in ACAD9 unbound vs. in complex with ECSIT$_{CTER}$. Source data are provided as a Source Data file.

essential for ECSIT binding and MCIA formation (Supplementary Videos 1 and 2).

Similarly to the ACAD9$_{WT}$-ECSIT$_{CTER}$ dataset, residues 455-485 could not be built into the ACAD9$_{S191A}$ map. However, increasing the threshold of the unsharpened ACAD9$_{WT}$ map reveals some density at the base of ACAD9$_{S191A}$ model, hinting at their possible position. Thus, we fitted the missing residues into this signal, using the coordinates of the ACAD9 AF2 model where residues 455-464 and 475-492 form flexible loops, sandwiching a short α-helix (res. 465-474). The lack of definition in the density points to a mobility of this region (Supplementary Fig. 7E) and may be explained by the lack of contacts between the α-helix and the ACAD9 core. Interestingly, we do not see signal for these residues in the ACAD9$_{S191A}$ dataset, probably due to a larger number of particles being included in the higher resolution reconstruction, leading to the signal being averaged out due to high mobility.

**ACAD9 has structural features unique among the ACAD family**

The comparison of our ACAD9$_{S191A}$ structure with both the VLCAD-based ACAD9 homology model[7] and a very recently published VLCAD crystal structure (PDB:7S7G[19]) revealed a high structural similarity, with

RMSD values of 1.16 Å and 1.11 Å, respectively (Fig. 4A). A visualisation of the evolutionary conserved residues also reveals that the dehydrogenase domains are quite conserved within the ACAD family[20] (Fig. 4B). Intriguingly, and in agreement with previous studies[7,12], ACAD9 is the only ACAD family member capable of binding ECSIT and assisting CI assembly (Supplementary Fig. 2A), whereas VLCAD does not fulfil these functions[12]. Thus, given that the ECSIT binding induces the flipping of the gatekeeper loop, we generated ACAD9 point mutants intended to mimic the VLCAD sequence on this loop and analysed them by DLS and mass photometry. In regards to their ECSIT-binding properties, these mutants behave essentially like ACAD9$_{WT}$ (Fig. 4C–F), suggesting that, although the conformational flexibility of the ACAD9 gatekeeper loop is important for the dimer stability and the enzymatic activity, it is nevertheless not decisive to confer its ability to bind ECSIT.

Therefore, in our quest to identify the distinct ECSIT binding features of ACAD9, we compared the FAD cofactor sites between our ACAD9 structure and the reported VLCAD structure (Fig. 5A, B and Supplementary Fig. 8A–C)[16]. In ACAD9, the FAD molecule is situated on top of the core strands and adopts an elongated conformation for

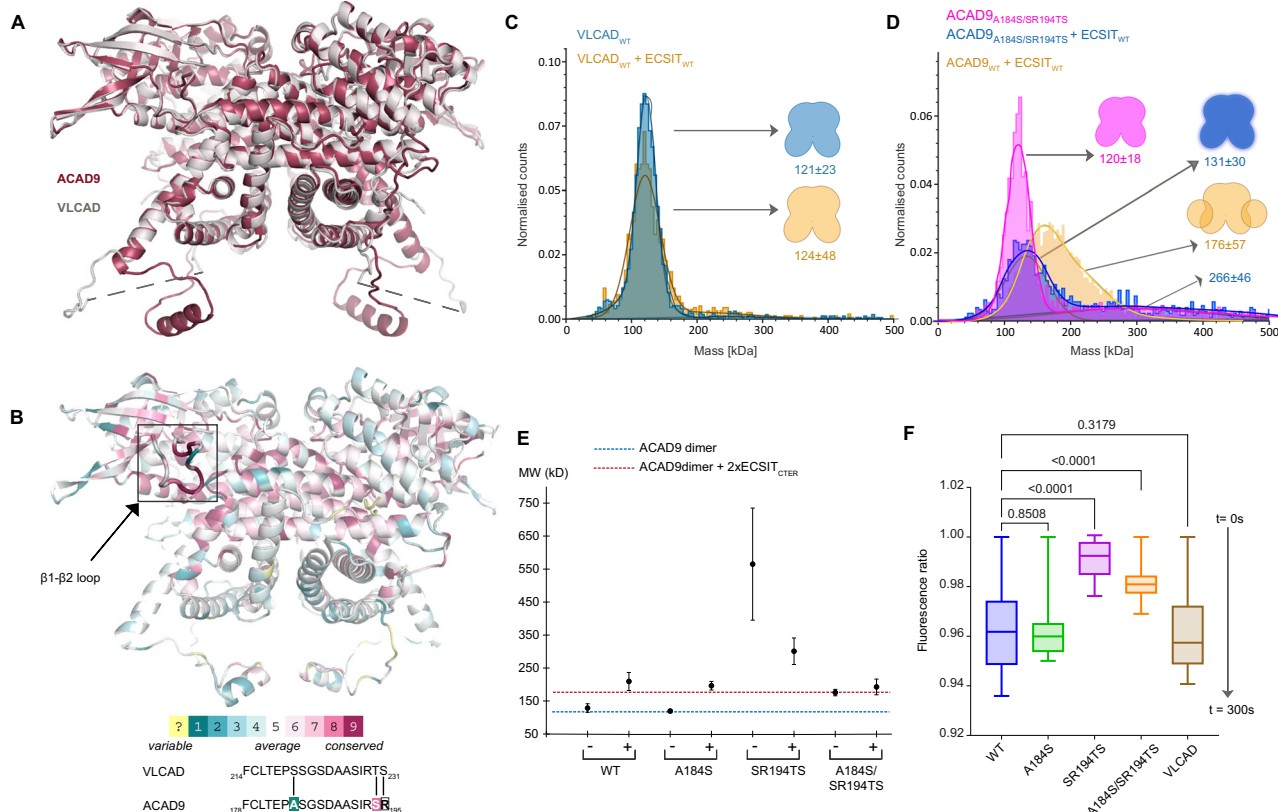

**Fig. 4 | The FAD-gatekeeper loop stabilises the ACAD dimeric conformation but does not determine ECSIT binding. A** Superposition of VLCAD crystal structure (7S7G) and the ACAD9$_{S191A}$ cryo-EM structure. **B** Representation of ACAD9 structure based on the evolutionary conserved residues estimated by the ConSurf prediction software[20]. Aligned sequences of the ACAD9 and VLCAD FAD-gatekeeper loop. Mutated residues are shown in bold. **C** VLCAD$_{WT}$ is a very stable dimer (blue) and does not form a complex when reconstituted with ECSIT$_{CTER}$ (orange). **D** ACAD9$_{A184S/SR194TS}$ primarily forms a dimer (pink) although it also forms higher order species upon reconstitution with ECSIT$_{CTER}$ (blue), indicating a decrease in the stability of the complex, while retaining the ability to bind to ECSIT. Experiments were repeated thrice with similar results. **E** DLS measurements of ACAD9 mutants designed to mimic the VLCAD loop. The ACAD9$_{A184S}$ mutant exhibits similar behaviour to ACAD9$_{WT}$. The double mutant ACAD9$_{SR194TS}$ is destabilised, although the effect was less profound in complex with ECSIT$_{CTER}$; some complex is formed but at higher MW than ACAD9$_{WT}$-ECSIT$_{CTER}$. In contrast,

the double mutant ACAD9$_{A184S/SR194TS}$ shows no change in MW before and after reconstitution with ECSIT$_{CTER}$, however, we can attribute this to a reduction in protein stability while retaining the ability to bind ECSIT$_{CTER}$. Blue and red dashed lines indicate the expected MW for ACAD9$_{WT}$ homodimer and ACAD9$_{WT}$-ECSIT$_{CTER}$ complex, respectively. Data are presented as mean values ± SD of at least $n = 3$ biological replicas. **F** Acyl-CoA dehydrogenase (ACAD) activity of ACAD9$_{WT}$, ACAD9 mutants, and VLCAD determined by an ETF fluorescence reduction assay. After addition of the ACAD specific substrate palmitoyl-CoA (C16:0), there is a clear loss of ETF fluorescence in ACAD9$_{WT}$, ACAD9$_{A184S}$ and VLCAD in comparison to the other ACAD9 mutants, over 300 s of reaction measurement. Data are presented as box-whisker plots with median quartiles and from minimum to maximum values, at least $n = 3$ biological replicas. Statistical significance was calculated with an RM-one way ANOVA with Dunnett's multiple comparisons test relative to ACAD9. $p$ values are indicated (non-significant $p = 0.8508$, $p = 0.3179$; ****$p \le 0.0001$). Source data are provided as a Source Data file.

maximal interaction with the FAD-binding domain (Supplementary Fig. 7D and Fig. 5B), very similar to the FAD positioning in VLCAD. The overall architecture of the pocket is analogous in both proteins. However, in ACAD9 the volume of the FAD binding pocket is larger and bonding interactions between the ACAD9 protein and the FAD molecule are longer and weaker (Supplementary Fig. 8A, C), suggesting that the fold of the FAD-binding domain and the cofactor conformation are not independent, whereby the arrangement of the domain determines the position and shape of the pocket and thus the cofactor conformation.

Interestingly, in VLCAD, two Methionine residues located on a 3$_{10}$-helix (res. 437–443) from the second protomer make sulfur-π interactions with a Tryptophan residue (W209) and the FAD isoalloxazine moiety of the first protomer, while in ACAD9 the corresponding Leucine and Threonine residues are unable to mediate similar interactions with the equivalent Tryptophan (W213) (Fig. 5A, B). We thus generated an ACAD9 mutant, ACAD9$_{VLCAD}$, to mimic VLCAD by replacing residues $^{401}$LGGLGYT$^{407}$ by the corresponding VLCAD residues $^{437}$MGGMGFM$^{443}$ (Fig. 5D, E). Although ACAD9$_{VLCAD}$ forms a stable

dimer, exhibiting a profile similar to VLCAD by mass photometry (Fig. 5D), it does not appear to form a complex with ECSIT$_{CTER}$. Furthermore, the presence of ECSIT$_{CTER}$ seems to induce dissociation of ACAD9$_{VLCAD}$ into monomers (Fig. 5D, E). On the other hand, although the counterpart VLCAD$_{ACAD9}$ mutant (Fig. 5C, E) appears to form a less stable dimer than the VLCAD$_{WT}$, it forms a complex with ECSIT$_{CTER}$, even if larger species also appear (Fig. 5C, E). Collectively, these findings suggest that in VLCAD, the dyad of Methionine residues located on a helix in the N-terminal α-helical domain (α-dom2) of the neighbouring protomer (as described in ref. 16) are responsible for stabilising the FAD in the binding pocket (Fig. 5A, B). This ability to retain the FAD cofactor is directly correlated with the stability of the homodimer but inversely correlated with the ability to bind ECSIT.

Lastly, another potential site that could account for the differences between ACAD9 and VLCAD in CI assembly lies on the 35-residue linker (res. 450-485 in ACAD9 and 486−521 in VLCAD), which shows the highest sequence divergence between the two proteins[7]. The recently reported VLCAD crystal structure models most of those residues (res. 486-499) facing away from the protein and projecting

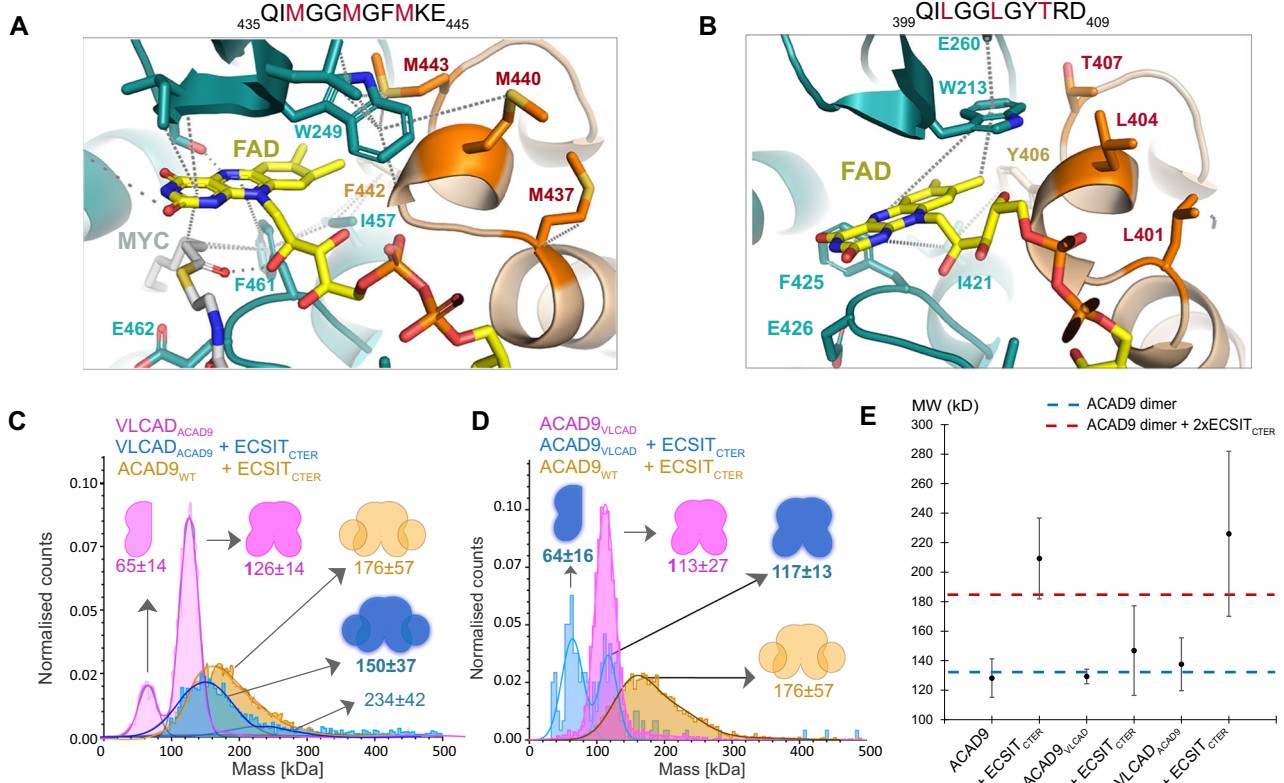

**Fig. 5 | Identification of the features that distinguishes ACAD9 from VLCAD.**
**A** Three Methionine residues (M437, M440, M443) are situated at the top of the VLCAD FAD site and contribute to the stability of the FAD molecule by M440 and M443 forming sulfur-π interactions with the isoalloxazine moiety and W209.
**B** Unlike VLCAD, ACAD9 has a Threonine and two Leucine residues (in red) and is unable to provide stability to the FAD molecule. **C** Mass photometry analysis shows that when VLCAD is mutated to mimic ACAD9 (VLCAD$_{ACAD9}$, pink), there is a reduction in stability compared to VLCAD$_{WT}$ (Fig. 4C). When reconstituted with ECSIT$_{CTER}$, there is a shift in the MW, indicating the formation of a VLCAD$_{ACAD9}$-ECSIT$_{CTER}$ complex (in blue). **D** The reverse mutation of ACAD9 to mimic VLCAD (ACAD9$_{VLCAD}$) produces a more stable dimer (pink). Upon reconstitution of ECSIT$_{CTER}$, we see an appearance of a monomeric ACAD9$_{VLCAD}$ (blue) indicating the destabilisation of the ACAD9 dimer and no significant shift to a higher MW,

demonstrating a stark reduction in the ability of ACAD9$_{VLCAD}$ to form a complex with ECSIT$_{CTER}$ in comparison to WT (orange). Experiments were repeated thrice with similar results. **E** DLS measurements of ACAD9$_{WT}$, ACAD9$_{VLCAD}$ and VLCAD$_{ACAD9}$ show similar behaviour. However, in the case of ACAD9$_{VLCAD}$, there is a significant reduction in complex formation in comparison to ACAD$_{WT}$-ECSIT$_{CTER}$, implying that this mutation does indeed evoke VLCAD-like tendencies. The reverse is also true in the case of VLCAD$_{ACAD9}$, a large average MW is seen, indicating the formation of a VLCAD$_{ACAD9}$-ECSIT$_{CTER}$ complex. This indicates that VLCAD$_{ACAD9}$ behaves less like VLCAD$_{WT}$ and more like ACAD9$_{WT}$ through the mutation of these key residues. Blue and red dashed lines indicate the expected MW for ACAD9 homodimer and ACAD9$_{WT}$-ECSIT$_{CTER}$ complex, respectively. Data are presented as mean values ± SD of at least $n = 3$ biological replicas. Source data are provided as a Source Data file.

outward[19]. This extended loop mediates *trans* interactions with the equivalent loop of another symmetry related VLCAD molecule (Supplementary Fig. 9A), stabilising the quaternary conformation of a dimer of homodimers[19]. However, superposition with the ACAD9$_{WT}$ structure reveals that in ACAD9 this loop tucks into the core of the same homodimer instead (Supplementary Fig. 9B). The higher conformational flexibility in VLCAD could be accounted by Gly488 at the end of α16 helix, which turns to be a hydrophobic residue (Ile452) in ACAD9 (Supplementary Fig. 9B). Excitingly, alignment with the ACAD9$_{WT}$-ECSIT$_{CTER}$ complex structure shows that residues of the VLCAD flexible loop are in fact positioned near the ECSIT binding site in the ACAD9$_{WT}$-ECSIT$_{CTER}$ complex (Fig. 1D and Supplementary Fig. 9C). Therefore, it appears that the equivalent ECSIT binding site on VLCAD can be obstructed by the loop from another monomer. In fact, a structural alignment on this region between VLCAD residues 486-498 and ECSIT residues 320–334 (reversed to match VLCAD sequence positioning, 334-320) shows some overlap in hydrophobic character (Supplementary Fig. 9D). In summary, these observations suggest that even if VLCAD contains an equivalent ECSIT binding site, it is unable to specifically bind ECSIT as it can be blocked by VLCAD itself. This auto-inhibitory feature would not be the case for ACAD9,

as the equivalent sequence of residues are dissimilar (Supplementary Fig. 9D), indicating another possible reason for the specificity of ACAD9 for ECSIT in comparison to VLCAD.

## ECSIT$_{CTER}$ is phosphorylated and has an impact on MCIA complex stability
In addition to its pivotal role in the MCIA complex, ECSIT was also described as providing a bridge between Toll-like receptor (TLR) signal transduction and downstream signalling kinases[9]. Previous studies have shown evidence of ECSIT ubiquitination by the E3 ubiquitin ligase TRAF6 in the cytoplasm, which leads to ECSIT enrichment at the mitochondrial periphery to generate mROS[21] and its translocation into the nucleus to regulate NF-κB activity[22]. Furthermore, another study reported a non-identified post-translationally modified form of ECSIT dependent on a MAPK kinase activity and apparently required for its role in the TLR pathway[9]. Therefore, to gain an insight into ECSIT regulatory mechanisms and given the frequent crosstalk between ubiquitination and phosphorylation, we investigated whether ECSIT undergoes phosphorylation in the presence of MAPK kinases and whether this has an impact on ACAD9-ECSIT subcomplex formation.

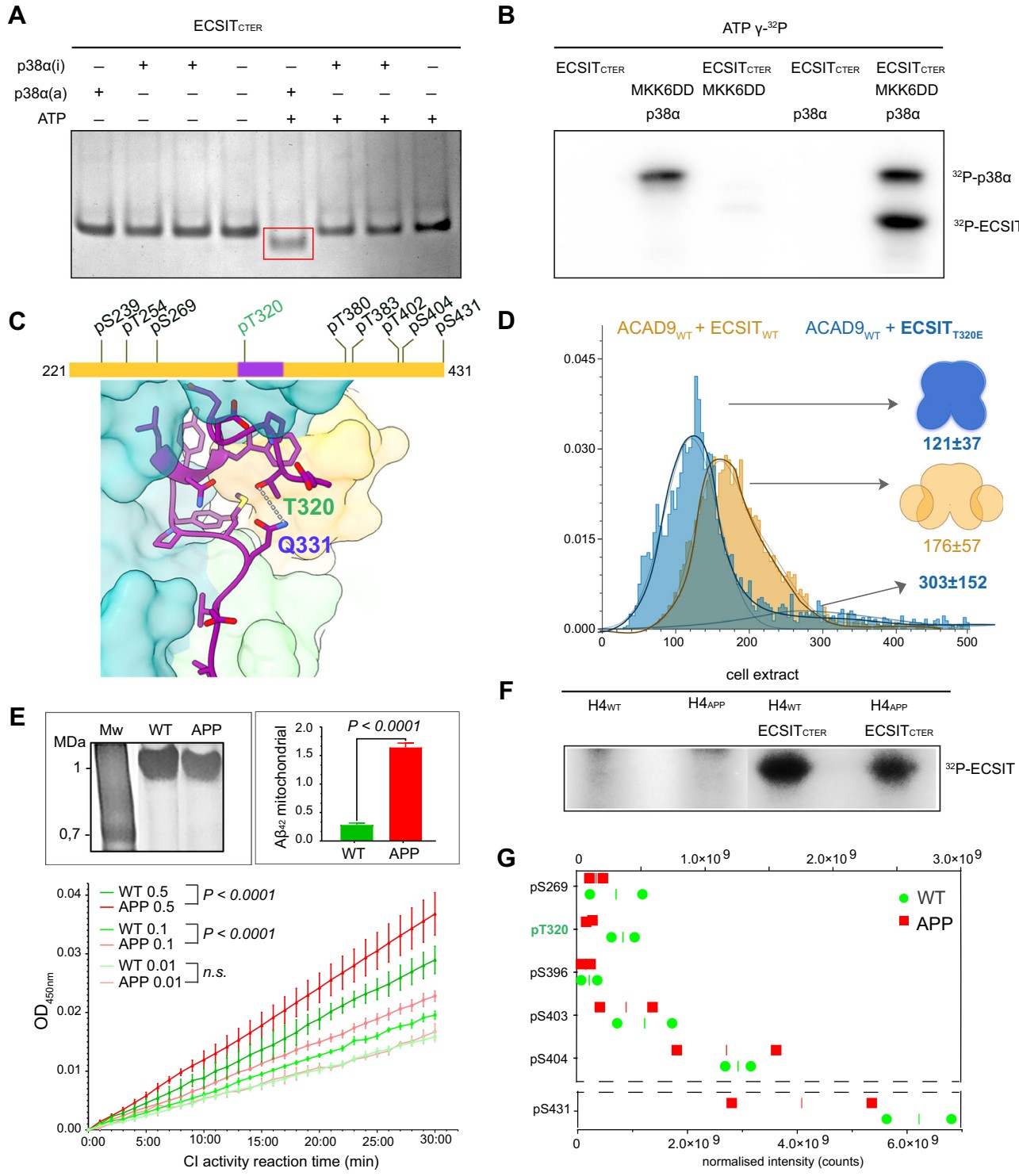

We first examined the potential phosphorylation of ECSIT in vitro by incubating ECSIT_CTER with a well-characterised MAPK kinase, p38α, activated in the response to ROS to function as a redox sensor[23]. Furthermore, p38α has been implicated in several neuronal functions that are relevant for brain physiology and has also been detected in the early stages of AD[23], making it a logical kinase candidate for our in vitro studies. Native gel electrophoresis showed that ECSIT_CTER migrates with higher mobility upon p38α kinase treatment (Fig. 6A), consistent with the gain of one or more negatively charged phosphate groups. Subsequent radiolabelling analysis confirmed the presence of strong ³²P-labelled bands in ECSIT_CTER treated with activated p38α, indicating

the direct phosphorylation of ECSIT by p38 MAPK kinase in vitro (Fig. 6B). Analysis of the faster migrating band by proteolytic in-gel digestion mass spectrometry revealed that ECSIT_CTER was multiphosphorylated (Supplementary Data 2). We were able to identify nine high-confidence Threonine/Serine phosphorylation sites, including a kinase recognition sequence TSS motif (res. 401–403)[24] (Fig. 6C).

Excitingly, one of the identified potential phosphorylation sites was Thr320, which resides within the ECSIT sequence visible in our ACAD9_WT-ECSIT_CTER structure (Fig. 6C). Thr320 forms hydrogen bonds with Gln331, stabilising the helical sequence that enters into the ACAD9 binding cavity. We therefore constructed an ECSIT_CTER

**Fig. 6 | Phosphorylation of ECSIT may regulate the binding affinity for ACAD9.**
**A** ECSIT treated with MKK6-activated p38α MAP kinase reveals a different mobility shift compared to the non-treated protein on a native gel. **B** Radioblotting assays of active P38α MAP kinase and ECSIT p38α-treated sample corroborate ECSIT phosphorylation. Experiments repeated twice with similar results. **C** Diagram of the phosphorylation sites identified by mass spectrometry. ECSIT$_{CTER}$ (res. 320–334) observed by cryo-EM is highlighted in purple and the Thr320 phosphosite in green. The interaction of ECSIT$_{CTER}$ Thr320 with Gln331 suggests that a Thr320 phosphate group would generate a steric clash resulting incompatible for ACAD9 binding. **D** Mass photometry reveals that the ECSIT$_{T320E}$ phosphomimetic mutant affects complex formation with ACAD9, resulting in large particles with the major species being an ACAD9 homodimer with unbound ECSIT. Experiments were repeated thrice with similar results. **E** Assays with H4 human neuroglioma cells both wild-type (WT) and overexpressing human amyloid precursor protein carrying Alzheimer's-related mutation KM670/671NL (APP). Top left, native gel showing

immunopurified fully assembled CI (1 MDa) from both cell types and subjected to the activity assays. Top right, Aβ$_{1-42}$ detection in mitochondria isolated from WT (green) and APP (red) cells by immunoassays, represented as mean values ± SD for $n = 3$ biological replicas. Statistical significance was calculated with unpaired $t$-test, ****$p < 0.0001$). Bottom, NADH-dehydrogenase activity assays of CI in WT (green) and APP (red) cells at different concentrations (0.5, 0.1, and 0.01 mg/ml, respectively). Data are presented as mean values ± SD of $n = 3$ biological replicas. Statistical significance was calculated with two-way ANOVA followed by Sidak's multiple comparison. Significant $p < 0.0001$. n.s. non-significant. **F** Cell extracts from H4 cells, both WT and APP, incubated with ECSIT$_{CTER}$ confirmed the phosphorylation of ECSIT$_{CTER}$ ex cellulo by a kinase pool. **G** Quantitative mapping of the ECSIT$_{CTER}$ phosphopeptides indicate a decreased phosphorylation level under amyloidogenic conditions, with Thr320 (in green) displaying the highest differential level. Data are presented as normalised values and median of $n = 2$ biological replicas. Source data are provided as a Source Data file and in Supplementary Data 2 and 3.

phosphomimetic mutant where Thr320 was replaced by a Glutamate (T320E). Remarkably, this mutant exhibited a significantly reduced binding affinity for ACAD9 (Fig. 6D and Supplementary Fig. 10A), suggesting that the phosphorylation of Thr320 may negatively regulate ACAD9 binding.

We next sought to examine the potential ECSIT$_{CTER}$ phosphorylation sites in vivo. To this end, we treated ECSIT$_{CTER}$ with an endogenous kinase pool from human neuronal cells (Fig. 6E, F). We observed strong $^{32}$P-radiolabelling of ECSIT$_{CTER}$, providing evidence that ECSIT undergoes phosphorylation ex cellulo. Subsequent analyses by proteolytic in-gel digestion mass spectrometry confirmed that ECSIT$_{CTER}$ was multiphosphorylated (Fig. 6G), and identified six high-confidence Threonine/Serine sites, including five already detected in vitro: Ser269, Thr320, the TSS motif and the C-terminal residue Ser431 (Fig. 6C, G and Supplementary Data 3).

### ECSIT$_{CTER}$ phosphorylation is affected by Aβ soluble oligomers
In parallel, we performed a quantitative phosphoproteomic study upon exposure of ECSIT$_{CTER}$ to soluble Aβ oligomers (Fig. 6E, F). Intraneuronal Aβ can translocate directly from the endoplasmic reticulum into mitochondria, where it may affect mitochondrial respiration[25]. Thus, we sought to investigate whether mitochondrial Aβ alters the phosphorylation levels of ECSIT. Overexpression of the amyloid precursor protein (APP) carrying the AD-related Swedish mutation (KM670/671NL) in neuroglioma cells enables the investigation of the toxic Aβ$_{1-42}$ soluble oligomeric form as may occur during AD[26,27] (Fig. 6E, right inset). Interestingly, while we obtained similar levels of immunopurified CI from both cell types (Fig. 6E, left inset), the amyloidogenic cells exhibited a significantly enhanced NADH-dehydrogenase activity (Fig. 6E, bottom), raising the possibility that soluble Aβ$_{1-42}$ oligomers lead to aberrant CI hyperactivity and could be a primary source of oxidative stress. Finally, we investigated whether the Aβ$_{1-42}$ soluble oligomers affect ECSIT$_{CTER}$ phosphorylation and found that the degree of phosphorylation of the six phosphorylation sites differed greatly between wild-type and amyloidogenic neuronal cells, with a clear tendency to decrease upon exposure to amyloids (Fig. 6G and Supplementary Data 3).

### Discussion
The Mitochondrial Complex I Assembly complex (MCIA) is required for the biogenesis of Complex I and is therefore crucial for the activation of the OXPHOS system[6]. FAO and OXPHOS are key pathways involved in cellular energetics[28]. Despite their functional relationship, evidence for a physical interaction between the two pathways is sparse[29]. Understanding how FAO and OXPHOS proteins interact and how defects in these two metabolic pathways contribute to mitochondrial disease pathogenesis is thus of critical importance for the development of new tailored therapeutic strategies.

Here, we provide high-resolution structural insights into the interface and assembly of the MCIA subcomplex ACAD9-ECSIT, with one molecule of ECSIT bound to each ACAD9 protomer at the crossing between the dehydrogenase and vestigial domains. ECSIT induces the flipping of the ACAD9 β1-β2 loop acting as gatekeeper by ~10 Å such as to allow the binding of an ECSIT segment (res. 320–334) and the release of the FAD cofactor from ACAD9 (Fig. 3 and Supplementary Videos 1 and 2). Once formed, the complex is very stable (Supplementary Figs. 1A–C and 2A). Based on the present data and our previously published low resolution information[12], we propose this stability may arise from ECSIT's ability to envelop ACAD9, made possible by long stretches of flexible loops (res. 306–320 and 335–356, Supplementary Fig. 6A, D).

Furthermore, the AF2-predicted ACAD9$_{WT}$-ECSIT$_{CTER}$ model supports our previous suggestions that ECSIT entwines around the vestigial domain of ACAD9 (Supplementary Fig. 6A, D)[12], in contrast to an alternative SAXS model placing the ECSIT binding interface at the N-terminus of ACAD9[13]. The fact that there is an overall low confidence in the position of the globular core of C-terminal ECSIT with respect to ACAD9 (Supplementary Fig. 6B) not only supports our experimental data but may also explain why the folded region of ECSIT is not visible in the ACAD9$_{WT}$-ECSIT$_{CTER}$ reconstruction: this region is probably highly mobile due to the flexibility of residues 306–320 and 335–356. Worthy of note, because AF2 is unable to predict conformational changes that occur during protein-protein interactions[30], while the AF2 model corroborates the experimentally determined ACAD9$_{WT}$-ECSIT$_{CTER}$ interface, it does not predict the displacement of the gatekeeper loop.

The experiments with the ECSIT-mimicking short peptide identified from the cryo-EM map show that this stretch of residues is key to the deflavination of ACAD9 (Supplementary Fig. 2B). Intriguingly, despite the high similarity between the ACAD9 and VLCAD structures, only ACAD9 holds the unique ability to bind ECSIT and participate in CI assembly[7]. In order to investigate key differential residues, we first examined the divergent residues on the gatekeeper loop. However, mutant analyses reveal that, even if the sequence of this loop has an impact in the acyl-CoA activity of the enzyme, the exact nature of the amino acids is not directly responsible for the formation of the complex with ECSIT (Fig. 4B–F). We then examined the FAD binding site in both protein structures[16] (Fig. 5A, B and Supplementary Fig. 8) to check whether the FAD binding stability is related to the ECSIT binding capacity. We observed that in fact, ACAD9 has a larger FAD binding pocket, leading to a reduced number of bonding interactions that are typically longer and therefore, weaker (Fig. 5A, B and Supplementary Fig. 8A–C). A less tightly bound cofactor is less stable and thus the feasibility of deflavination should be higher than if the FAD was tightly bound. Very interestingly, through the identification of VLCAD residues in the FAD binding, we have reversed the ACAD9 behaviour in the

counterpart mutant, achieving reduction both in ECSIT binding and in deflavination (Fig. 5B, D, E). In fact, the counterpart mutation in VLCAD also induces a reversed behaviour, facilitating ECSIT binding to VLCAD (Fig. 5A, C, E). These findings suggest that changes in hydrophobicity/π-bonding capabilities in the FAD binding pocket strongly affect the FAD stability and indirectly, the binding to ECSIT as well (Fig. 5A–E). In fact, it appears that ACAD9 is in an equilibrium between two states characterised by a high FAD binding affinity in its closed conformation and a low FAD binding affinity in its open conformation. ECSIT seems to exert a non-competitive allosteric influence by stabilising the low FAD binding affinity state, even though there are no apparent clashes between ECSIT and FAD. Furthermore, re-examination of a VLCAD structure at higher resolution and in absence of fatty acid substrate revealed a unique quaternary structure of VLCAD homodimers stabilised by an extended conformation of the ~35 residue stretch at the base of VLCAD (residues 486–521, Supplementary Fig. 9A) in contrast to the equivalent residues in ACAD9 (Supplementary Fig. 9B)[19]. Notably, this region is also where both proteins show the highest sequence divergence, which could further account for dissimilar conformational abilities. Remarkably, this extended loop occupies the equivalent ECSIT binding site (Supplementary Fig. 9C, D), suggesting that a reason why VLCAD is unable to bind ECSIT may be the obstruction of this binding site by VLCAD itself. Taken together, subtle differences in sequence are thus likely to play a critical role in impacting ACAD9-ACAD9 and VLCAD-ECSIT interactions, possibly accounting for the differences between VLCAD and ACAD9 in CI assembly participation, as previously suggested[7].

Our finding that ECSIT undergoes phosphorylation raises the possibility that a post-translational modification may provide an additional layer of regulation of MCIA complex assembly (Fig. 7). Although further investigation is required to determine the identity of the ECSIT kinases and the mechanism by which these might regulate MCIA complex formation, the identification of five high-confidence Threonine/Serine sites in $ECSIT_{CTER}$ suggests a potential role for phosphorylation in modulating the ACAD9-ECSIT interaction. In particular, our analyses of the phosphomimetic mutant T320E suggest that phosphorylation at Thr320, either solely or in combination with other phosphorylation sites, could conceivably induce conformational changes that would disrupt the fitting of the ECSIT segment in the ACAD9 cavity (Fig. 6C, D). Notably, analysis of the segment sequence across species reveals a high amino acid conservation except for Thr320, which seems to have evolved in higher primates from an Isoleucine (Supplementary Fig. 10B). Along this line, it is interesting to highlight that Threonine is an intermediate in Isoleucine biosynthetic pathway, which supports the hypothesis that rapid evolution of phosphorylation sites could provide a way to fit the environment by rewiring the regulation of signal pathways. Furthermore, previous studies have found evidence that the functional potential of phosphorylation sites are increased with their evolutionary age, especially for Serine and Threonine, in disordered regions[31].

Finally, our observations of decreased $ECSIT_{CTER}$ phosphorylation in response to Aβ oligomers, coupled to a significantly enhanced NADH-dehydrogenase CI activity, further suggests that ECSIT plays a role as a molecular link between mitochondrial bioenergetics and early amyloid pathology (Fig. 6E). Although these results are preliminary and warrant additional experimentation for a deeper understanding to determine how amyloids contribute to the functional activities of ECSIT, it seems that soluble Aβ would somehow favour ECSIT

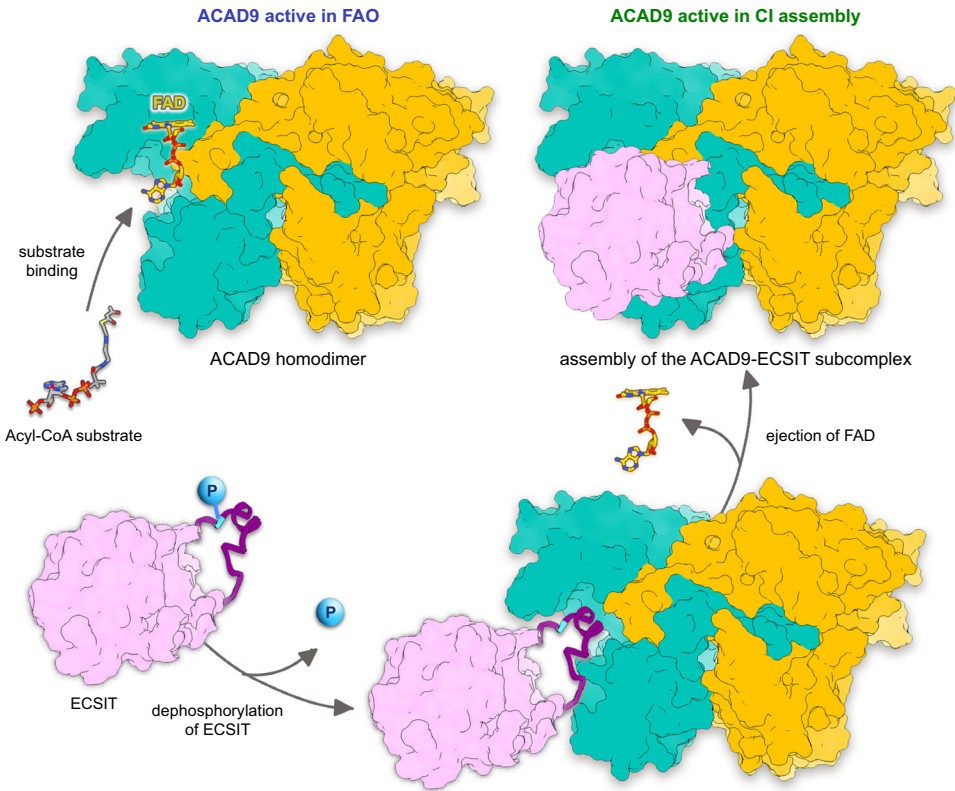

**Fig. 7 | Proposed model of the mechanism of ACAD9-ECSIT assembly and its functional implications in FAO and OXPHOS pathways.** In absence of ECSIT, ACAD9 acts as an acyl-CoA dehydrogenase enzyme in the first step of the FAO pathway. Dehydrogenation of the acyl-CoA substrate is concomitant with the reduction of the FAD prosthetic group into FADH₂. Upon recognition of ACAD9 by ECSIT, the gatekeeper loop in ACAD9 flips upwards, allowing the concomitant binding of ECSIT and deflavination, shutting down the dehydrogenase activity of ACAD9 and becoming committed to CI assembly. ECSIT dephosphorylation would enable the binding to ACAD9 and hence, act as a potential trigger to favour the assembly of the MCIA complex.

dephosphorylation, apparently required for the initial recognition of MCIA partners. Interestingly, some studies show that soluble oligomeric Aβ is sufficient to dramatically alter MAPK pathways before amyloid deposition[32]. Aβ can even undergo phosphorylation that seem to exert increased toxicity in human neurons as compared to other known Aβ species[33]. Therefore, ECSIT phosphorylation could either be directly regulated through downstream kinases or by the crosstalk with other signalling pathways. Under early amyloidogenic conditions, a stabilised MCIA complex would in turn contribute to a stabilised CI and thus, enhance its activity. Progressively, CI over-activity would lead to an NADH/NAD+ redox imbalance, generating oxidative stress and exacerbating the accumulation of Aβ oligomers in a vicious cycle, which ultimately may result in the inhibition of CI activity, in agreement with previous studies[3,21].

In summary, our study provides insights into the molecular mechanisms of ACAD9-ECSIT assembly and its functional implications in FAO and OXPHOS pathways, a process that may in turn be regulated by the phosphorylation of ECSIT (Fig. 7). Here, in the absence of ECSIT, ACAD9 acts as an acyl-CoA dehydrogenase enzyme in the first step of the FAO pathway. The recognition of aromatic residues on the ECSIT 320–334 residue segment by the ACAD9 gatekeeper loop would induce its opening, enabling the deflavination of ACAD9 and the binding of ECSIT. This dual mechanism seems to be unique to ACAD9 by virtue of the flexible nature of its FAD binding site and the compactness of the 35 amino acid stretch at the base of the molecule. Our findings that ECSIT can be phosphorylated and that this has a negative effect on its affinity for ACAD9 imply that an ECSIT (de-)phosphorylation would be the trigger of ECSIT binding to ACAD9 and hence, a regulatory switch of the MCIA complex assembly. The observations that Aβ soluble oligomers increase the activity of CI but also decrease ECSIT phosphorylation levels provide further evidence that ECSIT might be involved in AD pathogenesis[11]. By compromising both the assembly of CI and the regulation of FAO, ECSIT could act as a reprogrammer of OXPHOS and FAO pathways that may lead to altered metabolism in brain mitochondria and ultimately neurodegeneration[12].

## Methods

### DNA plasmids

The human DNA sequences coding for ECSIT (ECSIT$_{CTER}$, residues 221–431) and mature ACAD9 (residues 38–621) were *E. coli* codon-optimised and synthesised by ShineGene Molecular Biotech. ACADVL was a gift from Nicola Burgess-Brown (Addgene plasmid #38838). The DNA plasmids were constructed following the restriction free cloning[12]. ACAD9-related constructs were cloned into pET-21d(+) while VLCAD-related constructs were cloned into pNIC28-BSA. Mutant plasmids were generated using the QuikChange Lightning Site-Directed Mutagenesis Kit (Agilent). ACAD9-related constructs were cloned into pET21d(+) while VLCAD-related constructs were cloned into pNIC28-BSA. Details of all primers and plasmids used can be found in Supplementary Data 1.

### Recombinant protein expression and purification

ECSIT$_{CTER}$ (residues 221–431) and all ECSIT mutants (ECSIT$_{W324A}$, ECSIT$_{Y327F}$, ECSIT$_{Y327A}$, ECSIT$_{Y328F}$, ECSIT$_{Y328A}$, ECSIT$_{T320E}$) were expressed in *E. coli* BL21 Star (DE3) cells (ThermoFisher Scientific) in LB medium at 25 °C and subsequently induced with 250 μM isopropyl-β-galactopyranoside (IPTG). ACAD9$_{WT}$ (res. 38–621) and ACAD9 mutants (ACAD9$_{S191A}$, ACAD9$_{A184S}$, ACAD9$_{SR194TS}$, ACAD9$_{A184S/SR194TS}$, ACAD9$_{VLCAD}$) were expressed in *E. coli* C43 (DE3) cells (Lucigen) while VLCAD$_{WT}$ (res. 72–655) and VLCAD$_{ACAD9}$ in *E. coli* Rosetta 2 (DE3) cells (Merck), and cultured in Terrific Broth medium at 37 °C and induced with IPTG at a final concentration of 500 μM. All cells were harvested 18 h after induction. Bacterial pellets were resuspended in a lysis buffer (200 mM potassium phosphate, 300 mM or 250 mM NaCl, 0.25 mM EDTA, 1 mM DTT, 0.2% Tween-20, pH 7.5) supplemented with protease inhibitor cocktails (Merck) and DNase I (Merck). Bacterial lysis was performed by sonication (QSonica) followed by removal of cell debris by centrifugation for 25 min, 45,000 × g (Avanti J-20 XP centrifuge, Beckman Coulter) at 4 °C. Protein purification was conducted using a combination of affinity and size-exclusion chromatography (SEC) performed using Äkta Purification systems (GE Healthcare). Lysate containing the protein of interest was loaded onto a 5 mL HisTrap column (GE Healthcare) equilibrated in 200 mM potassium phosphate, 300 mM or 250 mM NaCl, 5 mM βME, pH 7.5 and eluted with a linear imidazole gradient reaching a final concentration of 500 mM imidazole in 200 mM potassium phosphate, 300 mM or 250 mM NaCl, 5 βME, pH 7.5. Protein-containing fractions were identified using sodium dodecyl sulfate-polyacrylamide gel electrophoresis (SDS-PAGE) and selected for further purification. VLCAD (and mutant) and ECSIT$_{CTER}$ (and mutants) were purified on Superdex 200 10/300 GL SEC column (GE Healthcare), equilibrated in 25 mM HEPES, 300 mM or 250 mM NaCl, 1 mM TCEP, pH 7.5. ACAD9$_{WT}$ (and mutants) were dialysed in SEC buffer. Sample purity was assessed using SDS-PAGE and concentrated using 30 kDa and 10 kDa cut-off Amicon Ultra filters for ACAD9/VLCAD and ECSIT, respectively. Single proteins were stored at −80 °C until use.

The MCIA subcomplex ACAD9-ECSIT (all variants) was assembled by warming an aliquot of ACAD9 to 32 °C and slowly adding ECSIT to a final molar ratio of 1:1.25. After ~5 min the protein mixture was plunged into ice and left for at least 15 min before use. Protein complex samples were used in analyses immediately after formation. Source data of SEC runs are provided as a Source Data file. Uncropped SDS-PAGE of reconstituted ACAD9$_{WT}$-ECSIT$_{CTER}$ is provided in the Source Data file and in the Supplementary Information.

Constitutively active MKK6DD (S207D/T211D mutant) and WT p38α were produced as described in ref. 34.

### Isothermal titration calorimetry (ITC)

Measurements were done on a Micro PEAQ-ITC (Malvern Panalytical) at 25 °C. ACAD9$_{WT}$, ECSIT$_{CTER}$, ECSIT$_{T320E}$ were dialysed overnight against 50 mM Tris-HCl pH 7.5, 250 mM NaCl. For binding studies, ACAD9$_{WT}$ at 49 μM was introduced into the sample cell and titrated with either ECSIT$_{CTER}$ or ECSIT$_{T320E}$ respectively at 300 μM over a period of 50 min. Data analysis was carried out using the MicroCal PEAQ-ITC Analysis Software (Malvern Panalytical v1.41). The experiments were performed thrice with similar results. Source data are provided as a Source Data file.

### Mass photometry data acquisition and analysis

Mass photometry experiments were carried out using a Refeyn OneMP Mass photometer system (Refeyn Ltd, Oxford, UK). All data were acquired with a 3*10 μm (at 1 kHz) size field of view. Refeyn AcquireMP 2.3.0 and Refeyn DiscoverMP 2.3.0 software packages were used to record movies and analyse data respectively using standard settings. Microscope coverslips (high precision glass coverslips, Marienfeld) were cleaned following the Refeyn Ltd Individual rinsing procedure. Reusable self-adhesive silicone culture wells (Grace Bio-Labs reusable CultureWell™ gaskets) were used to keep the sample droplet shape. Contrast-to-mass calibration was carried out using a mixture of native proteins (NativeMark Unstained Protein Standard, Invitrogen) with molecular weight (MW) of 66, 146, 480 and 1048 kDa. Immediately prior to the measurements, protein stocks were diluted directly in analysis buffer (25 mM HEPES pH 7.5, 250 mM NaCl), or in 4.8 μM of ECSIT solution in case of complex formation, to reach a concentration of 20–50 nM for ACAD9. To this end, 1 μl of protein solution was added into 19 μl of analysis buffer, to reach a final drop volume of 20 μl. The experiments were performed thrice with similar results. Source data are provided as a Source Data file.

## Size exclusion chromatography coupled to multi-angle laser light scattering (SEC-MALLS)

The molecular mass of $ECSIT_{CTER}$ and $ACAD9_{WT}$ in solution was determined by SEC coupled to multi-angle laser light scattering (SEC-MALLS) using a Superdex 200 10/300 GL column equilibrated in 25 mM HEPES, 250 mM NaCl, pH 7.8 buffer. The measurements were performed at 20 °C using 50 μl of proteins at 5 mg/ml with a flow rate of 0.5 ml/min and eluted proteins were monitored using a DAWN-EOS detector with a laser emitting at 690 nm for online MALLS measurement (Wyatt Technology) and with a RI2000 detector for online refractive index measurements (SchambeckSFD). Molecular mass calculations were performed with the ASTRA software using a refractive index increment (dn/dc) of 0.185 ml/g. Data were visualised using OriginPro 9.0 (OriginLab) software. The experiments were performed thrice with similar results. Source data are provided as a Source Data file.

## Dynamic light scattering (DLS)

DLS measurement were performed at 278 K on a Zetasizer Nano S (Malvern Panalytical), using 20 μl of protein sample at 1–2 mg/ml. A Micro Cell quartz cuvette was incubated at 4 °C until a stable temperature was reached. At least 3 biological replicates comprising from 2 to 4 technical replicates were measured for each sample. Data was processed using Zetasizer Software 8.01.4906. Data are presented as mean values ± SD. Source data are provided as a Source Data file.

## Acyl-CoA dehydrogenase (ACAD) activity assay

Anaerobic ETF fluorescence reduction assay was used for sensitive, accurate determination of ACAD9 and VLCAD activities. The assay was performed as previously described[12], with reaction volumes of 200 μl in Elplasia black-walled 96-well plates (Corning). Reactions were measured in an Infinite M200 PRO reader (TECAN) set to 32 °C, using Ex340/Em490. 30 data points were collected for each sample over a 1 min measurement window. Glucose oxidase (Sigma G2133, ~20 U/ml final concentration) and catalase (Sigma C30, 0.5 μl/mL final) were added and fluorescence was zeroed (Ex340/Em490). Then, enzyme sample (400 ng recombinant ACAD9, VLCAD or ACAD9 mutants respectively) and human ETF (2 μM final concentration) in buffer 50 mM Tris-HCl, 0.5% w/v glucose, pH 8.0 were added. Background fluorescence was recorded for 1 min. The reaction was initiated by addition of 30 μM final concentration of palmitoyl Acyl-CoA substrate (Sigma P9716) and immediately read for 300 s. Each experiment was performed in 3 to 4 replicates. Statistical significance was calculated with an RM-one way ANOVA with Dunnett's multiple comparisons test relative to ACAD9. $p$ values are indicated (non-significant $p = 0.8508$, $p = 0.3179$; ****$p \leq 0.0001$). Statistical analyses were carried out using GraphPad Prism version 10 (GraphPad Software). Source data are provided as a Source Data file.

## Cryo-EM sample preparation

$ACAD9_{WT}$-$ECSIT_{CTER}$ was purified through gel filtration as described above and stored in a buffer containing 25 mM HEPES, 300 mM NaCl and 1 mM TCEP, pH 7.5. The $ACAD9_{WT}$ and $ACAD9_{S191A}$ samples were dialysed into the same buffer and stored without gel filtration. For cryo-EM grid preparation, the samples were diluted in the same buffer to 0.17 mg/ml, 0.03 mg/l and 0.02 mg/ml respectively, applied to a glow-discharged R 1.2/1.3 300 mesh UltrAuFoil gold grid (Quantifoil Micro Tools GmbH) and plunge-frozen in liquid ethane using a Vitrobot Mark IV (Thermo Scientific) operated at 8 °C at 100% humidity.

## Cryo-EM data collection

Grids were pre-screened on a Glacios microscope (Thermo Scientific) of the EM platform of the Institute of Structural Biology (IBS), Grenoble, France. Datasets were collected at the European Synchrotron Radiation Facility CM01, on a Titan Krios microscope (Thermo

Scientific) operating at 300 kV with a Quantum energy filter (slit width 20 eV)[35]. The $ACAD9_{WT}$-$ECSIT_{CTER}$ and $ACAD9_{WT}$ data sets were recorded at zero degrees stage tilt on a K2 summit direct electron detector (Gatan) running in counting mode, at a magnification of ×165,000, corresponding to a pixel size of 0.827 Å/pixel at the specimen level. The $ACAD9_{S191A}$ data set was recorded at a 35° stage tilt, on a K3 direct electron detector (Gatan) operating in super-resolution mode, at a magnification of ×105,000, with a sampling pixel size of 0.42 Å/pixel, and was binned two-fold for data processing. For $ACAD9_{WT}$-$ECSIT_{CTER}$, a total of 8001 movies of 40 frames were acquired with a dose rate of 9.5 e-/Å²/s and a total exposure time of 3 s per frame, corresponding to a total dose of 41.7 e-/Å². For $ACAD9_{WT}$, a total of 7731 40 frames-movies were acquired with a dose rate of 6.7 e-/Å²/s and a total exposure time of 6 s per frame, corresponding to a total dose of 40.2 e-/Å². For $ACAD9_{S191A}$, a total of 2641 40 frames-movies were acquired with a dose rate of 14.7 e-/Å²/s and a total exposure time of 3 s per frame, corresponding to a total dose of 62.5 e-/Å². See Supplementary Table 1 for a summary of data collection information.

## Cryo-EM data processing

All image analysis was conducted using the CryoSPARC v3.2 software[36]. Imported movies were motion-corrected, dose weighted, and, in the case of the super-resolution $ACAD9_{S191A}$ data set, Fourier cropped (2x). All frames of the $ACAD9_{WT}$-$ECSIT_{CTER}$ and $ACAD9_{WT}$ movies were used, whereas the first 5 frames of the $ACAD9_{S191A}$ movies had to be discarded due to a stage drift during data acquisition at 35° tilt. Initial CTF estimation was performed on the aligned and dose-weighted summed frames. Micrographs were then manually screened, resulting in 7789 micrographs of $ACAD9_{WT}$-$ECSIT_{CTER}$, 7598 micrographs of $ACAD9_{WT}$ and 2608 micrographs of $ACAD9_{S191A}$ selected for further processing.

Supplementary Fig. 3 summarises the image processing workflows used for the three data sets. The $ACAD9_{WT}$-$ECSIT_{CTER}$ data set was the first to be acquired and processed. The first round of particle picking was performed automatically using the blob picker, resulting in ~2,200,000 picked particles. Particles were then extracted with a box size of 256 × 256 pixels and subjected to 2D classification, which immediately revealed very strongly predominant C2-symmetric top view orientation, similar to the situation encountered in our earlier work with a different ECSIT construct[12]. In total, ~1,600,000 particles corresponding to the best classes showing clear secondary structural features were selected for the generation of the first ab initio 3D volume which was then used as a reference for homogeneous refinement with applied C2 symmetry. Although the resulting 3D reconstruction had a nominal resolution of 3.07 Å, it was clearly anisotropic due to the strong preferential top view orientation and the underrepresentation of the side and tilted views. Thus, projections of this reconstruction according to the directions of rare views were selected as templates for the next round of particle picking, leading to ~2,500,000 picked particles. These were extracted and subjected to 2D classification resulting in a more diverse set of classes. Seven classes (~115,000 particles, Supplementary Fig. 5A) representing the most different views of the molecule were used for a homogenous 3D refinement with the previous reconstruction as a reference and a C2 symmetry imposed, which yielded a map at 3.23 Å. This improved but still visually somewhat elongated map was used to generate templates corresponding to different subsets of the rarest views for several further picking and 2D classification jobs performed in parallel. Particles corresponding to the rare views were combined, duplicate particles resulting from different picking jobs removed with a dedicated cryoSPARC routine, and classes representing the most different orientations carefully selected for a 3D refinement, resulting in a new 3D volume and leading to generation of new rare view templates. This procedure was iterated until no further improvement of the reconstructed 3D volume was detected. This 3D volume was then used as a reference for homogeneous 3D refinement of the corresponding

279,483 particles, followed by a non-uniform 3D refinement. This last refinement step resulted in a final 3D map with an estimated resolution of 3.07 Å according to the Fourier Shell Correlation (FSC = 0.143), which was then post-processed using a soft mask and sharpened with a B-factor of −171 Å$^2$ for visualisation and model building (Fig. 1B and Supplementary Fig. 5B–D).

The second acquired and processed data set, ACAD9$_{WT}$, suffered from an apparently even more severe preferential orientation. Therefore, after numerous trials with the blob picker and the templates obtained from the resulting 2D class averages, we resolved to use the rare views of the final ACAD9$_{WT}$-ECSIT$_{CTER}$ map filtered to 30 Å resolution for template picking. A total ~1,120,000 particles were picked in several parallel jobs and duplicates removed. The particles were extracted with a box size of 256 × 256 pixels and subjected to 2D classification. Homogenous 3D refinement with the low pass filtered ACAD9$_{WT}$-ECSIT$_{CTER}$ map as a reference and imposing the C2 symmetry was then performed on several different subsets of classes while carefully selecting rare views of reasonable quality. The best, although anisotropic, reconstruction that could be obtained by this approach included only 31,209 particles and had a nominal resolution of 3.91 Å. These particles were then subjected to a 3D classification resulting in two classes containing 17,922 and 13,287 particles, and corresponding 3D maps with estimated average resolutions of 4.71 Å and 5.98 Å, respectively. Visual assessment of these maps showed that the lower resolution map containing less particles was actually of higher quality and had a more isotropic resolution (Supplementary Fig. 5A, B, E, F). Homogeneous 3D refinement of this map showed a further improvement in its quality, leading to a final 3D reconstruction of ACAD9$_{WT}$ with an estimated average resolution of 4.5 Å (FSC = 0.143). This map was post-processed using a soft mask and sharpened with a B-factor of −180 Å$^2$.

Considering the difficulties encountered due to a very strongly preferred orientation of ACAD9$_{WT}$, we opted for a recording of the ACAD9$_{S191A}$ data set at a 35 degrees stage tilt in order to increase the proportion of the rare views. As for ACAD9$_{WT}$, the rare views of the low pass filtered ACAD9$_{WT}$-ECSIT$_{CTER}$ map were used for template picking. A total ~286,000 particles were picked, extracted with a box size of 256 × 256 pixels and subjected to 2D classification. The best classes containing a total of ~149,000 particles were selected (Supplementary Fig. 5A) and subjected to a homogenous 3D refinement, using the low pass filtered ACAD9$_{WT}$-ECSIT$_{CTER}$ map as a reference and imposing the C2 symmetry. The obtained volume had a nominal resolution of 4.02 Å but, similarly to what we observed for the first map of ACAD9$_{WT}$-ECSIT$_{CTER}$, clearly anisotropic. Addition of ~14,000 more particles corresponding to other classes resembling the rarer views and a subsequent 3D refinement resulted in a map of 3.92 Å resolution. A non-uniform 3D refinement of these ~163,000 particles gave a map with an estimated resolution of 3.63 Å (FSC = 0.143). A dip in the FSC curve at ~6.5 Å was however still indicative of a remaining anisotropy (Supplementary Fig. 5B). In addition, a visual inspection of the map showed that some regions, in particular the β-sheets were not resolvable. However, side chain information could be identified in most α-helices. Further data processing with additional particles and changing parameters showed no improvement in the map quality (Supplementary Fig. 5G, H). Thus, this latest 3D reconstruction of ACAD9$_{S191A}$ was manually sharpened with an input B factor of −180 Å$^2$ to allow better visualisation of the higher resolution features such as side chains and the orientation of the FAD cofactor.

## Model building and refinement

To build a model of ACAD9$_{WT}$-ECSIT$_{CTER}$, a homology model of ACAD9 dimer based on the human very long-chain Acyl-CoA dehydrogenase (VLCAD, PDB:3b96), was used as the initial structure[7]. AF2 models were not available at the time. In this model, each ACAD9 protomer contains residues 38–621 and binds one FAD molecule (a total of two FADs in

the ACAD9 dimer). The homology model was rigid-body fitted into the 3D reconstruction of ACAD9$_{WT}$-ECSIT$_{CTER}$ using ChimeraX 1.2[37]. A first round of real space refinement was conducted enabling rigid body, global minimisation, local grid search and atomic displacement parameter (ADP) refinement parameters; rotamer Ramachandran, non-crystallographic symmetry (NCS) and reference model (VLCAD-based homology model) restraints were also imposed. The output model was then manually corrected in Coot (v0.9.4.1)[38] followed by iterative real space refinements with PHENIX (v1.18.2)[39], using the same settings but removing the rigid body refinement parameter.

Regions of the ACAD9 dimer model with no clearly corresponding cryo-EM density (454–486, FAD) were removed from the final model. Upon model building, additional density that could not be attributed to ACAD9 was observed in a pocket adjacent to the ACAD9 dehydrogenase domain on each protomer. We tentatively attributed this density to ECSIT and manually inserted short polyalanine chains into each density using Coot (v0.9.4.1)[38]. Each chain was seen to adopt a short α-helix composed of 8–10 residues, sandwiched between two loops. A total of 15 Alanine residues could be fitted into the visible density. Due to the high resolution of the cryo-EM map in this region, it was possible to identify characteristics of several aromatic side chains, in particular a Phenylalanine and two neighbouring Tyrosine residues, indicating a sequence WXYY where X represented an unidentified residue. Comparison with the sequence of the ECSIT construct enabled us to detect a short amino acid sequence, totalling 15 amino acids corresponding to residues 320–334, with aromatic side chain positions matching those observed in the ACAD9$_{WT}$-ECSIT$_{CTER}$ cryo-EM map. The sequence was then mutated in Coot (v0.9.4.1)[38] in agreement with the identified ECSIT sequence. The final ACAD9$_{WT}$-ECSIT$_{CTER}$ model was validated using the comprehensive validation tool in PHENIX (v1.18.2)[39] (Supplementary Fig. 4A).

Considering that the final cryo-EM map of ACAD9$_{WT}$ was at 4.5 Å resolution and showed high levels of anisotropy, no attempts were made to build an atomic model of ACAD9$_{WT}$ in the absence of ECSIT. Thus, a simple rigid body fit of the ACAD9$_{WT}$-ECSIT$_{CTER}$ model and the AF2 ACAD9 model (res. 38–621) was performed in ChimeraX 1.2[37] (Supplementary Fig. 4B).

Model building into the ACAD9$_{S191A}$ cryo-EM map was done using both the VLCAD-based ACAD9 homology model and the final ACAD9$_{WT}$-ECSIT$_{CTER}$ models in parallel. Rigid body fitting in ChimeraX 1.2 revealed that the ACAD9$_{WT}$-ECSIT$_{CTER}$ model fitted better into the ACAD9$_{S191A}$ cryo-EM map than the VLCAD-based ACAD9 homology model. Therefore, the ACAD9$_{WT}$-ECSIT$_{CTER}$ model was used for further structure refinement. As demonstrated in our earlier work[12], the FAD is not present in the ACAD9$_{WT}$-ECSIT$_{CTER}$ cryo-EM map. Therefore, in the resulting atomic model, the coordinates of the FAD molecules were taken from the aligned PDB of the homology model. Similarly to ACAD9$_{WT}$-ECSIT$_{CTER}$, the ACAD9$_{S191A}$ cryo-EM map does not display density for the flexible helices located next to the vestigial domain of ACAD9 (residues 454–485). As a result, these were removed from the model. Real space refinement of secondary structural features as well as side chains was iteratively conducted in Coot (v0.9.4.1) followed by rigid body refinement using PHENIX (v1.18.2)[39]. The final ACAD9$_{S191A}$ model was also validated using the comprehensive validation tool in PHENIX (v1.18.2)[39] (Supplementary Fig. 4C). A summary of refinement and model validation statistics can be found in Supplementary Table 1. Visualisation of structures and diagrams to analyse ACAD9, ECSIT and FAD interactions were done with ChimeraX 1.2[37], PyMOL v4.60[40] and LigPlot+ v.2.2.8[41].

## AlphaFold modelling

Structures of ACAD9 (human, residues 1–621, monomer) and ECSIT (human, residues 1–431) were available in the AlphaFold Protein Structure Database (AF2 v2.2.4)[17,30]. The PDB files were downloaded and compared to the structures produced from experimental data. For

the structural comparison of ECSIT model (residues 320–334) built based on the cryo-EM map, the AF2 ECSIT model was cropped to match the sequence of our construct without the addition of His-Tags. The structures were aligned and RMSD values generated using ChimeraX 1.2[37].

AlphaFold modelling of the ACAD9-ECSIT subcomplex was conducted using ColabFold (v1.3.0) prediction tools[42]. The protein sequences for ACAD9 (human) and ECSIT (human) were taken from UniPROT[43] and modified to match the constructs used in our experiments: ACAD9 contained residues 38–621 and ECSIT contained residues 221–431. Structure prediction of the ACAD9-ECSIT subcomplex was performed using two ACAD9 sequences and two ECSIT sequences, based on information obtained from our experimental results.

We took the best predicted model from the five proposed models for our analysis. The Predicted Aligned Error (PAE) plot generated alongside the docked model showed low values (<15), representing well defined relative positions and orientations (Supplementary Fig. 6A), for the majority of the residues in the ACAD9 dimer, the exception being for residues 455-486 which have high PAE values (~30). Overall PAE values for ECSIT are high, however lower values are shown for the residues corresponding to those modelled in the $ACAD9_{WT}$-$ECSIT_{CTER}$ model, where the PAE value is low. Additionally, alignment of the ECSIT residues built based on the cryo-EM map with the calculated model shows a very high agreement (RMSD of 1.045 Å) (Supplementary Fig. 6C), providing further support to our experimental results. Supplementary Fig. 6 describing the AF results was generated using ChimeraX 1.2[37].

## Mammalian cell culturing

Human neuroglioma H4 cells, both wild-type (WT) and stably transfected with human amyloid precursor protein (APP) carrying the AD-related Swedish mutation (KM670/671NL)[44], were kindly provided by Dr. Patrick Aloy, Institute for Research in Biomedicine, Barcelona, and derived from the ATCC catalogue (https://www.atcc.org/products/htb-148). Cells were cultured in Dulbecco's Modified Eagle Medium (DMEM)-low glucose and supplemented with GlutaMAX, pyruvate (Thermo Fisher), 10%(v/v) fetal bovine serum (FBS, GIBCO) and 100 U/ml penicillin–streptomycin (Invitrogen) at 37 °C and 5% $CO_2$ in a humidified incubator. For mitochondrial assays, cells were trypsinized at 80% confluence after 72-h incubation on T225 flasks ($10^6$ cells/flask). For $ECSIT_{CTER}$ phosphorylation assays, cells were cultured in galactose-free media and trypsinized at 80 % confluence after 4-h incubation on T225 flasks ($10^6$ cells/flask).

## Mitochondria isolation, CI immunopurification, CI activity assay and mitochondrial $A\beta_{1-42}$ detection

About $25 \times 10^6$ cells were resuspended in homogenisation buffer (10 mM Tris-HCl pH 6.7, 10 mM KCl, 0.15 mM $MgCl_2$, 1 mM PMSF, 1 mM DTT) and then transferred to a glass homogeniser and incubated for 10 min on ice. Cells were lysed using a tight-fitting pestle for ~2 min. The homogenised cellular extract was then gently mixed with 2 M Sucrose and centrifuged three times at $1200 \times g$ for 5 min to obtain the supernatant. Mitochondria were isolated by differential centrifugation steps according to published protocols[45]. Mitochondria were pelleted by centrifugation at $7000 \times g$ for 10 min and centrifuged twice in wash buffer (250 mM sucrose, 10 mM Tris-HCl, pH 6.7, 0.15 mM $MgCl_2$, 1 mM PMSF and 1 mM DTT) at $9500 \times g$ for 5 min. Complex I (CI) was immunopurified from H4 cell mitochondria using a commercial kit (Abcam ab109721) according to the manufacturer instructions. All steps were carried out at 4 °C. Briefly, mitochondria were solubilised with 1% n-dodecyl β-D-maltoside (DDM), centrifuged to remove insoluble material and incubated with the affinity beads overnight. The beads were washed twice before CI was eluted in buffer containing 200 mM glycine (pH 2.5) and 0.5% DDM. The pH was neutralised by addition of Tris base. Solubilised mitochondria were resuspended in

PBS buffer containing protease inhibitors (cOmpleteTM, Sigma) and concentration adjusted to 5.5 mg/ml using Bradford assay (BioRad). 1% DDM was then added, the preparation incubated on ice for 30 min and centrifuged at $16,000 \times g$ for 10 min. Samples at three different concentrations (0.01, 0.1 and 0.5 mg/ml, respectively) were added to the microplate wells precoated with a specific CI capture antibody. CI activity was analysed by measuring the absorbance at OD 450 nm in a kinetic mode at room temperature for up to 8 h in a microplate reader Quantamaster QM4CW (Horiba).

Every assay was carried out with three independent experiments and presented as a mean average with the standard deviation (sd). Statistically significant differences were determined by two-way ANOVA followed by Sidak's multiple comparison post-test to identify pair wise differences. Differences were considered significant at $p < 0.0001$. n.s.: non-significant.

The human Amyloid Beta (residues 1–42, $A\beta_{1-42}$) content in isolated mitochondria from WT and APP cells was measured three independent experiments at 9 mg/ml by enzyme-linked immunosorbent assay (ELISA) according to the manufacturer instructions (Thermo-Fisher khb3544). Statistically significant differences were determined by paired t-test to identify pair wise differences. Differences were considered significant at $p < 0.001$.

Statistical analyses were carried out using GraphPad Prism version 10 (GraphPad Software).

## Phosphorylation assays in vitro

P38α MAP kinase was activated with the active (DD) MKK6 kinase form following a similar procedure as in ref. 34. Protein samples were prepared on ice (MKK6DD:p38α:ECSIT) in 10 μl phosphorylation reaction buffer. For radiolabelling experiments, radiolabelled nucleotide ATP γ-$^{32}$P (Hartmann Analytic GmbH)) was diluted 1:10 into a 1 mM cold ATP solution. The phosphorylation reaction was started by adding 1 μl of nucleotide to each sample and incubated for 20 min at 30 °C. The reaction was stopped by adding 4 μl of SDS sample buffer (0.4% bromophenol blue, 0.4 M DTT, 0.2 M Tris pH 6.8, 8% SDS, 40% glycerol) and boiling for 5 min at 95 °C. Samples were centrifuged and loaded on Pre-cast 4–20% gradient Tris-Glycine gels (Thermo Fisher Scientific), run in Tris-Glycine running buffer (2.5 mM Tris Base, 19.2 mM glycine pH 8.3, 1% SDS) at 220 V for 40 min. The gels were exposed to a storage phosphor screens (GE Healthcare) overnight and imaged using a Typhoon scanner (GE Healthcare). Gels were then stained with InstantBlue (Expedeon) and scanned. Images were analysed using Image Lab (Bio-Rad) software (v2.3.0.07). For mass spectrometry analyses, we carried out the same protocol but using non-radioactive ATP. The samples were directly loaded on a 12% SDS PAGE to be analysed on a mass spectrometer as described below. Experiments repeated twice with similar results. Uncropped PAGE gels of ECSIT treated with MKK6-activated p38α MAP kinase from Fig. 6A–C, respectively, are provided as a Source Data file.

## Phosphorylation assays using cell extracts

About $50 \times 10^6$ cells cultured in glucose-free media and supplemented with 10 mM Galactose for 4 h were resuspended in kinase buffer (10 mM Tris-HCl, pH 6.7, 10 mM KCl, 10 mM $MgCl_2$, 1 mM DTT, 1% DDM). We then added 50 μl of purified $ECSIT_{CTER}$ at 4 mg/ml, 10 μl of phosphatase/protease inhibitors (Protease and Phosphatase Inhibitor Cocktail, EDTA-free, Abcam) and 5 mM ATP, all processed on ice. The incubation samples were then sonicated for 2 s and kept on ice for 5 min. We repeated this process twice. The samples were then incubated at 30 °C for 20 min to start the phosphorylation reaction and the reaction stopped by adding 50 μl of 4X SDS PAGE loading buffer (0.4% bromophenol blue, 0.4 M DTT, 0.2 M Tris pH 6.8, 8% SDS, 40% glycerol) to each sample. The samples were directly loaded on a 12% SDS PAGE to be analysed on a mass spectrometer as described below. For radiolabelling experiments, samples were

prepared similarly by adding the radiolabelled nucleotide after the sonication steps and processed as described above. Experiments repeated twice with similar results. Uncropped SDS-PAGE gels of ECSIT treated with cell extracts from Fig. 6F, G, respectively, are provided as a Source Data file.

## Mass spectrometry in-gel digestion

The mass spectrometry analyses were carried out at the Proteomics Core Facility from the EMBL Heidelberg. Bands corresponding to the protein of interest were cut from the gel and in-gel digestion with trypsin (Promega) was performed essentially as described in ref. 46. Uncropped SDS-PAGE gels of digested bands are provided as a Source Data file (Fig. 6C, G). Peptide extraction was done by sonication for 15 min, followed by centrifugation and supernatant collection. A solution of 50:50 water: acetonitrile, 1% formic acid was used for a second extraction. The supernatants of both extractions were pooled and dried in a vacuum concentrator. Peptides were dissolved in 10 μl of reconstitution buffer (96:4 water: acetonitrile, 1% formic acid) and analysed by LC-MS/MS. Experiments repeated twice with similar results.

## Mass spectrometry measurements

For LC-MS/MS measurements, an Orbitrap Fusion Lumos instrument (Thermo) coupled to an UltiMate 3000 RSLC nano LC system (Dionex) was used. Peptides were concentrated on a trapping cartridge (μ-Pre-column C18 PepMap 100, 5 μm, 300 μm i.d. x 5 mm, 100 Å) with a constant flow of 0.05% trifluoroacetic acid in water at 30 μl/min for 6 min. Subsequently, peptides were eluted and separated on the analytical column (nanoEase™ M/Z HSS T3 column 75 μm × 250 mm C18, 1.8 μm, 100 Å, Waters) using a gradient composed of Solvent A (3% DMSO, 0.1% formic acid in water) and solvent B (3% DMSO, 0.1% formic acid in acetonitrile) with a constant flow of 0.3 μl/min. The percentage of solvent B was stepwise increased from 2% to 6% in 6 min, to 24% for a further 41 min, to 40% in another 5 min and to 80% in 4 min. The outlet of the analytical column was coupled directly to the mass spectrometer using the nanoFlex source equipped with a Pico-Tip Emitter 360 μm OD × 20 μm ID; 10 μm tip (MS Vil). Instrument parameters: spray voltage of 2.4 kV; positive mode; capillary temperature 275 °C; Mass range 375–1650 $m/z$ (Full scan) in profile mode in the Orbitrap with resolution of 120,000; Fill time 50 ms with a limitation of 4e5 ions. Data dependent acquisition (DDA) mode, MS/MS scans were acquired in the Orbitrap with a resolution of 30,000, with a fill time of up to 86 ms and a limitation of 2e5 ions (AGC target). A normalised collision energy of 34 was applied (HCD). MS2 data were acquired in profile mode. Source data are provided as a Source Data file and in Supplementary Data files 2 and 3.

## Mass spectrometry data processing

The raw mass spectrometry data was processed with MaxQuant (v1.6.17.0)[47] and searched against the Uniprot-proteome UP000000625 database (E.coli, 4450 entries, May 2022, ECSIT in vitro experiment) or the Uniprot-proteome UP000005640 database (*homo sapiens*, 20603 entries, May 2022, ECSIT *ex cellulo* experiment) containing each the sequence of the protein of interest and common contaminants. The data was searched with the following modifications: Carbamidomethyl (C) as fixed modification, acetylation (Protein N-term), oxidation (M) and phosphorylation (STY) as variable modifications. The default mass error tolerance for the full scan MS spectra (20 ppm) and for MS/MS spectra (0.5 Da) was used. A maximum number of 3 missed cleavages was allowed. For protein identification, a minimum of 2 unique peptides with a peptide length of at least seven amino acids and a false discovery rate below 0.01 were required on the peptide and protein level. Match between runs was used with default parameters. The msms.txt output file of MaxQuant was used for MS1 Filtering with Skyline (v21.1.0.278)[48].

We applied the LFQ (label-free quantification) intensities for sample normalisation to best represent the ratio changes among different samples. Source data are provided as a Source Data file and in Supplementary Data files 2 and 3.

## Statistics and reproducibility

Data were collected on independent experiments. Statistics details are presented in the Source Data file, in the "Methods" section and in the figure legends where appropriate.

## Reporting summary

Further information on research design is available in the Nature Portfolio Reporting Summary linked to this article.

## Data availability

The mass spectrometry proteomics data have been deposited to the ProteomeXchange Consortium via the PRIDE partner repository[49] with the dataset identifiers PXD042858 (in vitro experiment) and PXD042905 (ECSIT experiment with cell extracts). The cryo-EM maps have been deposited in the Electron Microscopy Data Bank (EMDB) under accession codes EMD-17659 (ACAD9-WT in complex with ECSIT-CTER); EMD-17660 (Cryo-EM structure of human ACAD9-S191A); and EMD-17661 (ACAD9 homodimer WT). The atomic coordinates have been deposited in the Protein Data Bank (PDB) under accession codes PDB 8PHE (ACAD9-WT in complex with ECSIT-CTER) and PDB 8PHF (Cryo-EM structure of human ACAD9-S191A). Additional atomic coordinates referred to within this paper are in the PDB under the accession codes 7S7G (Crystal Structure Analysis of Human VLCAD) and 3B96 (Structural Basis for Substrate Fatty-Acyl Chain Specificity: Crystal Structure of Human Very-Long-Chain Acyl-CoA Dehydrogenase). Source data are provided with this paper.

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

## Acknowledgements

We are grateful to Dr. C. Mas (IBSG Grenoble) for support with ITC and mass photometry. We thank Dr. G. Leonard (ESRF), Dr. S. Bohic (ESRF, INSERM) and Dr. C. Petosa (IBS Grenoble) for discussions. We thank Dr. N. Burgess-Brown (Oxford University) for the VLCAD plasmid and Dr. P. Aloy (IRB Barcelona) for H4 cell lines. We are grateful to Dr. G. Schoehn for establishing and managing the IBS cryo-electron microscopy platform, and to Dr. E. Zarkadas for assistance at the Glacios microscope. This work used the EMBL Proteomics Core Facility and the platforms of the Grenoble Instruct Center (ISBG; UMS 3518 CNRS-CEA-UJF-EMBL) with support from FRISBI (ANR-10-INSB-05-02) and GRAL (ANR-10-LABX-49-01) within the Grenoble Partnership for Structural Biology (PSB). The IBS EM facility is supported by the Rhône-Alpes Region, the Fondation Recherche Medicale (FRM), the fonds FEDER and the GIS-Infrastructures

en Biologie Sante et Agronomie (IBISA). We acknowledge the ESRF for provision of beam time on CM01. The ESRF in-house Research Program supported this work. The EM work was funded by the European Union's Horizon 2020 research and innovation programme under grant agreement No. 647784 to I.G. This work was partially funded by the Agence Nationale de la Recherche (ANR) MitoCompAs (ANR-22-CE11-0031) program to I.G. and M.S.-L.

## Author contributions

M.S.-L. conceived and supervised the overall project in collaboration with I.G. for the EM analysis. M.S.-L. designed the constructs. L.M., S.A. and M.S. performed molecular biology and biochemistry, including sample preparation for all experiments. L.M., M.B.-V. and I.G. prepared EM grids and performed cryo-EM screening. L.M., E.K. and I.G. performed cryo-EM data collection. L.M., A.D. and I.G. performed cryo-EM data processing and structure refinement. L.M. and M.S.-L. performed structure analyses. J.J.S. performed the mass spectrometry analyses. P.J., J.v.V., M.W.B. and A.A.M. performed the AlphaFold modelling and analysis. S.A. and P.J. performed phosphorylation analyses. L.M. and M.S.-L. wrote the manuscript with significant contributions from I.G. All the authors revised the final manuscript.

## Competing interests

The authors declare no competing interests.
