## [Peer Review File · Nature Communications]

The assembly of the Mitochondrial Complex I Assembly complex uncovers a redox pathway coordinationReviewers' Comments:

Reviewer #1:

Remarks to the Author:

The manuscript describes structural and functional studies of a protein complex between ACAD9 protein and a C-terminal peptide from ECSIT protein. The two proteins are components of the Mitochondrial Complex I Assembly (MCIA) complex which is a transient complex during the assembly of mitochondrial complex I.

The study reports cryo-EM structures of dimeric ACAD9 in apo form wild type and S191A mutant as well as the structure of ACAD9 dimer with 2 bound C-terminal peptides of ECSIT protein. Structural modeling and analysis of low-resolution reconstructions of ACAD9-ECSIT_cteminal domain are consistent with the cryo-EM models. The high-resolution structures explain why the FAD co-factor dissociates when ECSIT C-terminal peptide binds. The authors performed extensive experiments to study the determinants of the complex formation using mass photometry and dynamic light scattering (DLS).

Then they asked a question about what determines the binding specificity of ECSIT for ACAD9 versus homologous VLCAD protein. They showed that exchanging loop that stabilizes FAD cofactor between ACAD9 and VLCAD swaps specificity between two proteins, which I found to be quite a remarkable achievement.

Moreover, the authors show that ECSIT C-terminal peptide can be phosphorylated and the phosphorylation results in reduced binding.

Finally, experiments in cells accumulating amyloidogenic peptides Ab1-42 were performed that displayed lower levels of phosphorylation of ECSIT C-terminal peptide. Intriguingly, the activity assays on immunoprecipitated complex I showed increased activity while the quantity remained unchanged when compared to control cells. This would suggest a direct interaction of Ab1-42 with complex I and does not seem to be linked to phosphorylation levels of ECSIT C-terminal peptide.

Overall, the manuscript contributes to the understanding of the properties and specificity determinants of the MCIA subcomplex and suggests new experiments to study complex I assembly.

Below I provide a few suggestions and list questions answering which will, in my opinion, improve the manuscript.

Major:

- 1) The rationale for using the C-terminal peptide instead of the complete ECSIT subunit is not clear and has to be explained.
- 2) In the mass spectrometry plots, the peak originating from ACAD9-ECSITCTER is not symmetric and appears to be broad. Do the authors have an explanation for this?
- 3) Line 93. Why Trp324 was chosen to test the effect of hydrogen bonds on peptide binding? E232 is also involved in hydrogen bonding. Additionally, W-> A mutation is quite a dramatic replacement. The choice should be explained.
- 4) Line 140. It is suggested that the b1-b2 loop prevents the dissociation of FAD, but in the end, no experimental evidence confirms the stabilizing role of the loop. Probably its role in stabilization should not be overstated. Just a suggestion: please don't use the word 'eject' in relation to FAD. It can be more useful to think of the dissociation of FAD caused by competition for the overlapping binding site with the ECSIT peptide. In this respect, it will be useful to mention whether and how FAD and ECSIT peptide overlap with each other.
- 5) I was wondering how can the b1-b2 loop create the barrier if it does not interact with FAD?
- 6) It may help readers if the authors could discuss the implications of their structure for understanding the structure and function of the complete MCIA complex. Can the complete MCIA be modeled with AF2?

7) Could the authors speculate on why amyloids change phosphorylation and how they may affect the activity of complex I?

Minor:

1) I could not find the source organism for the reported structures either in the results or in the Methods. Please add it.

2) Figure S2, please add FSC model-map curves.

3) Figure S3, please add a panel with AF2 models colored by chain, otherwise, it is impossible to distinguish the protein chains.

4) Figure 4 Labels on X axis are difficult to read.

5) From Figure 4B it seems that the b1-b2 loop is very conserved. If it is so, could you discuss the possible reasons for this?

6) Figure 5 legend for panel A. The residue numbers are shifted by -40.

7) Figure 6G please label X-axis.

8) Methods section. Please add versions of the software used, including in cryo-EM-related sections.

9) The models and maps should be deposited to EMDB and PDB.

Reviewer #2:

Remarks to the Author:

In this manuscript the authors expand on their prior work of identifying ECSIT as a regulator of ACAD9 activity as an agent of fatty acid beta oxidation or as a part of the mitochondrial Complex I assembly complex. The study uses a combination of structural analysis (with cryo-electron microscopy) and biochemical and molecular analysis. They identify a residue of ECSIT that binds within the pocket of ACAD9, thus competing with FAD binding. They also identify differences between ACAD9 and VLCAD which facilitate ECSIT binding and/or FAD binding. Among these differences, they describe differences in the binding pocket of ACAD9 vs VLCAD, differences in a loop structure that may act as a gatekeeper for FAD binding. Interestingly, they describe residues of ECSIT that can get phosphorylated in vitro by kinases at various residues, one of which is within the sequence of the 15 residue peptide that they are studying. Upon phosphorylation, ECSIT binding is decreased, which, based on their hypothesis, may be the switch to regulate ACAD9 function in FAO or in OXPHOS. Finally, they suggest that soluble amyloid beta leads to increased ECSIT phosphorylation in vitro, which may increase the proportion of ACAD9 involved in Complex I assembly and shift towards OXPHOS.

The investigators provide a significant amount of detail and insight about the binding of ECSIT with ACAD9, some based on alpha-fold modeling. The proposed regulatory mechanism by which ECSIT binding to ACAD9 ejects FAD from the FAD binding site to convert ACAD9 from an FAO enzyme to a member of the mitochondrial complex I assembly complex. And, that this may be regulated by the phosphorylation status of ECSIT. There is also some evidence to suggest that amyloid beta may impact ECSIT binding to ACAD9 which would have implications for Alzheimer's Disease, and may help to identify methods of developing Alzheimer's Disease treatments.

Although the authors have carried out an extensive set of experiments, many of the conclusions are not definitive. Some of the potentially important predictions are based on in vitro analysis without in vivo evidence. It is also unclear if the findings represent a sufficiently significant advance over their previous paper. I also have a number of questions and comments as listed below.

Comments:

- If there is an abundance of FAD present, can it eject ECSIT from its binding site on ACAD9?
 - What would be a proposed mechanism for how excess amyloid beta would cause decreased phosphorylation of ECSIT?
 - Does the reduced affinity between phospho-ECSIT and ACAD9 (in figure 6) lead to greater FAD binding in the FAD binding site of ACAD9?
 - Is there evidence for ACAD9 being phosphorylated within the ECSIT binding domain, leading to a similar disruption of the hydrogen bonds between Gln331 and T320 that is observed when the phosphomimetic was made for ECSIT?
 - It was unclear to me if the choice of kinases used for the study was adequately justified e.g. MAPK p38 alpha. Would one see similar results with a different MAP kinase? Is there any genetic evidence of change in FAO or OXPHOS in cells with p38 MAPK KO (e.g. with CRISPR)?
- While modeling, a portion of ECSIT (the C terminal residues 221-431) is used to say that there is a conformational change in ACAD9 that ejects FAD in the beta 1 and beta 2 loops adjacent to the FAD binding site. Since this is a portion of ECSIT, is it possible that the conformational change can occur because the rest of the globular ECSIT protein is not present to cause steric hindrance or another phenomenon to prevent this conformational change observed with the C terminal residues?

Other Comments:

- What are the loading controls for the phosphorylation assays?
- Why is supplemental figure 6 a supplemental figure rather than a main figure?

- Where are supplemental tables 2 and 3?
- Why doesn't one observe endogenous ECSIT phosphorylation in figure 6F?
- Were any statistics for the phosphorylation measurements done in figure 6?
- What is the sample size/n for the biological experiments?

Point-by-point responses to the Reviewers' comments

We are grateful to the Reviewers for their thorough revision and helpful comments, which have very much improved the quality of the manuscript. In the light of these we have revised our submission as follows:

We have clarified the relevance of our findings to provide a wide range of conclusions from the integrative approach we have used in this research.

We have added the FSC model-map curves in Figure S2 as requested. Commensurate with this Figure S2 has been split into **Figure S2** and **S3** for clarity purposes. For consistency with the text, former Figures S3 to S6 have been renumbered as new **Figures S4 to S7** respectively.

We describe in more detail the role of the β 1- β 2 loop in the overall mechanism of ACAD9-ECSIT interaction and the unique features in ACAD9 in comparison with VLCAD. We have produced 2 additional videos (supplementary **Video S1** and **S2**), included as additional material, to show the equivalent location in the ACAD9_{S191A} model with the ECSIT peptide in order to directly compare the difference of the atomic positions of the β 1- β 2 loop in open (Video S1) or in closed conformation (Video S2).

We have also clarified the rationale of our phosphorylation experiments and described in more detail the followed procedures in the methods. Furthermore, we included a manual inspection analysis in the *ex celullo* mass spectrometry data to be consistent with the in vitro data analysis, which has strengthen our conclusions (results in **Figure 6G**). Following reviewer #2 suggestion, we have now included supplementary Figure S6A into main **Figure 6** to complement the other. Furthermore, new Supplementary **Figure S7** contains the Coomassie-stained gels of the phosphorylation assays and the gel filtration chromatogram of ACAD9 in complex with the ECSIT phosphomimetic mutant T320E to show the decreased affinity of the complex as compared to the wild-type shown in **Figure S1A**.

As requested by the reviewers, we provide some molecular hypotheses about how amyloid-beta can alter the phosphorylation levels of ECSIT and the functional implications for the respiratory Complex I.

Following reviewer #1 suggestion, we have replaced “the ejection of FAD” by the more appropriate term “deflavination” throughout the manuscript.

EM data have been deposited to the EMDB and the structures of ACAD9-ECSIT_{CTER} and ACAD9_{S191A} to the PDB. The mass spectrometry data has been deposited to the ProteomeXchange Consortium via the PRIDE partner repository.

We have increased the font of labels of **Figure 4**. We have inserted AF2 models colored by chain in **Figure S5**. We have corrected **Figure 5** legend and add label X-axis to **Figure 6G**.

We have carefully read the text with the aim to detect and correct typographical or grammatical errors that we might have overlooked in the original manuscript.

For the sake of clarity, all the insertions and modifications, including – where appropriate – more recent references, are highlighted in yellow throughout the manuscript.

Please, find below specific answers to Reviewers' remarks.

Reviewer #1: The manuscript describes structural and functional studies of a protein complex between ACAD9 protein and a C-terminal peptide from ECSIT protein. The two proteins are components of the Mitochondrial Complex I Assembly (MCIA) complex which is a transient complex during the assembly of mitochondrial complex I.

The study reports cryo-EM structures of dimeric ACAD9 in apo form wild type and S191A mutant as well as the structure of ACAD9 dimer with 2 bound C-terminal peptides of ECSIT protein. Structural modeling and analysis of low-resolution reconstructions of ACAD9-ECSIT_cteminal domain are consistent with the cryo-EM models. The high-resolution structures explain why the FAD co-factor dissociates when ECSIT C-terminal peptide binds. The authors performed extensive experiments to study the determinants of the complex formation using mass photometry and dynamic light scattering (DLS).

Then they asked a question about what determines the binding specificity of ECSIT for ACAD9 versus homologous VLCAD protein. They showed that exchanging loop that stabilizes FAD cofactor between ACAD9 and VLCAD swaps specificity between two proteins, which I found to be quite a remarkable achievement.

Moreover, the authors show that ECSIT C-terminal peptide can be phosphorylated and the phosphorylation results in reduced binding.

Finally, experiments in cells accumulating amyloidogenic peptides Ab1-42 were performed that displayed lower levels of phosphorylation of ECSIT C-terminal peptide. Intriguingly, the activity assays on immunoprecipitated complex I showed increased activity while the quantity remained unchanged when compared to control cells. This would suggest a direct interaction of Ab1-42 with complex I and does not seem to be linked to phosphorylation levels of ECSIT C-terminal peptide.

Overall, the manuscript contributes to the understanding of the properties and specificity determinants of the MCIA subcomplex and suggests new experiments to study complex I assembly.

We thank reviewer #1 for the positive feedback regarding our manuscript. We have taken on board the comments and suggestions and we hope that this clarifies some of the points that were considered unclear before.

Below I provide a few suggestions and list questions answering which will, in my opinion, improve the manuscript.

Major:

1) The rationale for using the C-terminal peptide instead of the complete ECSIT subunit is not clear and has to be explained.

The results obtained in our previous work (Giachin et al. 2021, PMID: 33320993) indicated that the complete ECSIT subunit formed soluble aggregates when recombinantly purified from bacteria. In comparison, a C-terminal ECSIT fragment was identified using the ESPRIT technology as being soluble. In addition, previous yeast two-hybrid and BiFC assays determined that there was no interaction between the N-terminal domain of ECSIT and ACAD9, leading to us focusing on both a more stable and functionally relevant isoform of ECSIT, the C-terminal domain. As this was discussed in detail in the Giachin et al., with emphasis made on the reasons behind the choice of the C-terminal ECSIT particularly in the previous low resolution cryo-EM reconstruction, and to avoid repetition, explaining this was omitted from this manuscript but we specify now in the introduction that it is C-terminal ECSIT binding the trigger of ACAD9 deflavination (page 3).

2) In the mass spectrometry plots, the peak originating from ACAD9-ECSIT_{CTER} is not symmetric and appears to be broad. Do the authors have an explanation for this?

We presume the reviewer refers to the mass photometry plots. The appearance of a broad peak is due to the fact that even though the ECSIT_{CTER} protein is more stable than the previous C-terminal construct used in Giachin et al. (named ECSIT_{CTD}), it still has a tendency to form oligomeric species. As the mass photometry data is used for comparison between ACAD9_{WT}-ECSIT_{CTER} as the ideal subcomplex and the tested mutants, this does not affect the validity of the results. In addition, we have structurally proven that the reconstitution of ACAD9_{WT} and ECSIT_{CTER} results in the formation of a stable subcomplex despite the presence of additional ECSIT oligomeric species. We now describe the presence of oligomeric ECSIT species in the legend of Figure S1.

3) Line 93. Why Trp324 was chosen to test the effect of hydrogen bonds on peptide binding? E232 is also involved in hydrogen bonding. Additionally, W-> A mutation is quite a dramatic replacement. The choice should be explained.

To evaluate the effects of particular residues on the bonding interaction between ACAD9 and ECSIT, we focused our efforts on residues either located on the ACAD9 β 1- β 2 loop that undergoes the large conformational change or ECSIT residues buried deep into the binding pocket. Despite the fact that Glu323 is indeed a hydrogen bonding partner, the location of Trp324 and its relation to Ser191 on the β 1- β 2 loop suggested that Trp324 was a more likely candidate to contribute to the hydrogen bonding/ stability of the ACAD9-ECSIT interface. Referring to Figure 1E in the manuscript, the Glu323 sidechain is also not extremely well defined, indicating a degree of flexibility that would be unlikely in a strongly hydrogen bonded residue, whereas the position of Trp324 was clear. We postulated that the W324A mutant was a more sensible choice as in the use of, e.g. a W324F mutant, it would not be clear if the ACAD9-ECSIT subcomplex formation would be affected by the removal of the hydrogen bonding properties or the presence of a 6-

membered ring/larger aromatic system in the place of the pyrrole ring. We have included the rationale for selecting the W324A mutant in the text (page 5).

4) Line 140. It is suggested that the β 1- β 2 loop prevents the dissociation of FAD, but in the end, no experimental evidence confirms the stabilizing role of the loop. Probably its role in stabilization should not be overstated. Just a suggestion: please don't use the word 'eject' in relation to FAD. It can be more useful to think of the dissociation of FAD caused by competition for the overlapping binding site with the ECSIT peptide. In this respect, it will be useful to mention whether and how FAD and ECSIT peptide overlap with each other.

In order to provide a level of consistency when describing the phenomenon of the loss of FAD from ACAD9 upon ECSIT binding, we have continued to use the phrase 'eject' as was previously described in Giachin et al. However, it may be more appropriate to use 'deflavination' as was also used in Giachin et al. and in a similar publication Xia et al. 2021 (PMID: 3464699). This change has been applied throughout the article.

We take on board the comment made concerning the β 1- β 2 loop, however, we believe it is important to draw attention to the steric relevance of this loop due to its proximity to both the ECSIT peptide and FAD cofactor. We describe a large conformational change, however, we then clarify its surprisingly minor role by a range of biophysical measurements which we think will be of interest to readers. We believe a lack of extensive clarification would be confusing for readers due to the obvious nature of the loop displacement. Nevertheless, the reviewer is correct that, curiously, the FAD cofactor and the ECSIT peptide do not overlap. However, there is an overlap of atoms in the ECSIT peptide and the β 1- β 2 loop. Specifically, residues Tyr327 and Glu323 of ECSIT overlap directly with Ser187 and Ala189 of ACAD9, respectively. In addition, comparison of the two structures show a clear exposure of the FAD cofactor to the surrounding solvent when modelling the FAD cofactor from ACAD9S191A with the ACAD9 model from the ACAD9-ECSIT subcomplex.

We also do not mention in detail the contribution from bonds between each monomer of ACAD9. There is a 2.9Å hydrogen bond present between the hydroxyl oxygen of Asp188 of the β 1- β 2 loop and the amide nitrogen of Phe332 of the neighbouring monomer (loop α 11- α 12). We believe this contributes to the stability of both the homodimer and the FAD pocket. We have included this description in the text (page 6).

In the loop flipping mechanism that occurs upon ECSIT binding, this bond is broken and the stability of both the homodimer and the FAD pocket is reduced. This implies that although we show that no individual residue on the β 1- β 2 loop is responsible for the ECSIT recognition, its role in the overall mechanism of ACAD9-ECSIT interaction must be highlighted. Additionally, an interesting comparison can be made between VLCAD, which possesses several more bonding interactions between the β 1- β 2 loop and the adjacent monomer than in ACAD9, implying an increased stability. This was not discussed in detail. However, due to the difficulty in visualising this from 2D diagrams, we have produced 2 additional videos are included as additional material. Supplementary_video_S1 shows a close up of the ACAD9-ECSIT interaction site with relevant sidechains displayed; the FAD molecule from the aligned ACAD9S191A model included for reference. Supplementary_video_S2 shows the equivalent location in the ACAD9S191A model

with the ECSIT peptide in order to directly compare the difference of the atomic positions between the two conformations.

5) I was wondering how can the β 1- β 2 loop create the barrier if it does not interact with FAD?

We hope that reviewer #1 is satisfied with the response to this point that we believe we have covered in the response to point 4.

6) It may help readers if the authors could discuss the implications of their structure for understanding the structure and function of the complete MCIA complex. Can the complete MCIA be modeled with AF2?

In determining the structure of the ACAD9-ECSIT_{CTER} subcomplex, we have shown how ECSIT binds to ACAD9 and which residues are necessary for the deflavination of ACAD9 switching its ability from FAO to OXPHOS. The fact that post translational modifications also play a role in this regulatory mechanism, may also be important in the overall formation of the MCIA complex. We have found that ECSIT contains several phosphorylation sites, paving the way for further research on whether these play a role in the interactions with NDUFAF1 in the overall MCIA complex, with a different scope to that of this manuscript. With regards to the AF2 modelling of the full complex, we would stress that we have used AF2 as a complementary technique to our experimental data. Without experimental data, and the current limited capabilities of AF2 e.g. the inability to predict conformational changes in protein-protein interactions that we were able to visualise in our data, we believe that modelling the full MCIA complex is outside of the scope of this manuscript and would require appropriate experimental evidence to be interpreted correctly.

7) Could the authors speculate on why amyloids change phosphorylation and how they may affect the activity of complex I?

As we mention below in response to a comment from reviewer#2 regarding potential mechanisms for how excess amyloid beta would cause decreased phosphorylation of ECSIT, the effects of A β on the phosphorylation status of ECSIT can be very diverse due to the complexity of the amyloidogenic conditions. Furthermore, A β can even undergo phosphorylation and phosphorylated A β oligomers seem to exert increased toxicity in human neurons as compared to other known A β species (Rezaei-Ghaleh et al. 2016, PMID: 27072999).

Based on our findings, we can speculate that A β can exert an indirect activation of CI through the reverse phosphorylation of ECSIT in low amyloidogenic conditions, i.e., an accumulation of 4-fold A β with respect to healthy conditions. The activation of the respiratory chain can be one of the first activated pathways to generate energy to fight detrimental conditions, although CI over-activity would lead to an NADH/NAD⁺ redox imbalance, generating oxidative stress and exacerbating the accumulation of A β oligomers in a vicious cycle, which ultimately may result in the inhibition of CI activity.

We have included some of these speculations in the text, although, because of the highly speculative nature of these proposals, we believe it may form the basis for further study, with a broader scope of that of this publication.

Minor:

1) I could not find the source organism for the reported structures either in the results or in the Methods. Please add it.

We describe in the Materials and Methods that the DNA plasmids were constructed with the same protocol as Giachin et al., where it is stated that the source organism is human. We have added this information into this Materials and Methods for clarity.

2) Figure S2, please add FSC model-map curves.

We thank the reviewer for suggesting this addition. We have now added this information to Figure S2 and have split the figure into Figure S2 and Figure S3 for the sake of clarity.

3) Figure S3, please add a panel with AF2 models colored by chain, otherwise, it is impossible to distinguish the protein chains.

We thank reviewer #1 for this useful suggestion. We have included a reference figure of the AF2 model with the chains coloured in the same manner as is consistent throughout the rest of the manuscript. Of note, the new figure is Figure S4.

4) Figure 4 Labels on X axis are difficult to read.

We have taken this comment onboard and have increased the font size of the Figure 4 axis labels.

5) From Figure 4B it seems that the β 1- β 2 loop is very conserved. If it is so, could you discuss the possible reasons for this?

We suggest that the β 1- β 2 loop in both ACAD9 and VLCAD is important for providing a barrier between the adenosine monophosphate of the FAD molecule and the surrounding solvent. This is understandably necessary in both proteins in order to maintain the stability of the cofactor, enabling the dehydrogenase activity. Additionally, the presence of several interactions between the β 1- β 2 loop and the adjacent monomer are important for the homodimer stability. It is interesting to note that despite the fact that the β 1- β 2 loop is conserved between ACAD9 and VLCAD, there are more bonding interactions present between the VLCAD monomers than the ACAD9 monomers. This is likely to be linked to the difference in size of the FAD pocket, which we also suggest in our manuscript is directly linked to the FAD stability and the proteins ability to bind ECSIT. Additionally, as we describe in detail, the loop is not the discriminating factor for ECSIT binding, it is a larger concerted effort from many key aspects across the protein (other residues, FAD stability) so therefore it is not surprising that the sequences of the β 1- β 2 loop are quite conserved.

6) Figure 5 legend for panel A. The residue numbers are shifted by -40.

We thank reviewer #1 for bringing this to our attention. The residue numbers have been corrected.

7) Figure 6G please label X-axis.

We again thank reviewer #1 for noticing this slight oversight. The legend for the X-axis in Figure 6G has now been added.

8) Methods section. Please add versions of the software used, including in cryo-EM-related sections.

We thank reviewer #1 for this useful suggestion. We have now included the versions of software used throughout the Materials and Methods section in order to improve the information available to our readers.

9) The models and maps should be deposited to EMDB and PDB.

We have now submitted all three maps (ACAD9-ECSIT_{CTER}, ACAD9_{WT} and ACAD9_{S191A}) to the EMDB and the structures of ACAD9-ECSIT_{CTER} and ACAD9_{S191A} to the PDB.

Reviewer #2: In this manuscript the authors expand on their prior work of identifying ECSIT as a regulator of ACAD9 activity as an agent of fatty acid beta oxidation or as a part of the mitochondrial Complex I assembly complex. The study uses a combination of structural analysis (with cryo-electron microscopy) and biochemical and molecular analysis. They identify a residue of ECSIT that binds within the pocket of ACAD9, thus competing with FAD binding. They also identify differences between ACAD9 and VLCAD which facilitate ECSIT binding and/or FAD binding. Among these differences, they describe differences in the binding pocket of ACAD9 vs VLCAD, differences in a loop structure that may act as a gatekeeper for FAD binding. Interestingly, they describe residues of ECSIT that can get phosphorylated in vitro by kinases at various residues, one of which is within the sequence of the 15 residue peptide that they are studying.

Upon phosphorylation, ECSIT binding is decreased, which, based on their hypothesis, may be the switch to regulate ACAD9 function in FAO or in OXPHOS.

Finally, they suggest that soluble amyloid beta leads to increased ECSIT phosphorylation in vitro, which may increase the proportion of ACAD9 involved in Complex I assembly and shift towards OXPHOS.

We wish to clarify this statement that in fact we report that the presence of amyloidogenic conditions leads to a decreased phosphorylation of ECSIT, thus enabling an increased participation in the MCIA complex. We hope that reviewer #2 is satisfied with the responses to this point that we provide throughout this letter and in the revised manuscript.

The investigators provide a significant amount of detail and insight about the binding of ECSIT with ACAD9, some based on alpha-fold modelling. The proposed regulatory mechanism by which ECSIT binding to ACAD9 ejects FAD from the FAD binding site to convert ACAD9 from an FAO enzyme to a member of the mitochondrial complex I assembly complex. And, that this may be regulated by the phosphorylation status of ECSIT. There is also some evidence to suggest that amyloid beta may impact ECSIT binding to ACAD9 which would have implications for Alzheimer's Disease, and may help to identify methods of developing Alzheimer's Disease treatments.

Although the authors have carried out an extensive set of experiments, many of the conclusions are not definitive. Some of the potentially important predictions are based on in vitro analysis without in vivo evidence.

We thank reviewer #2 for recognising the huge amount of work that has gone into this manuscript. However, we do believe that we have been able to provide a wide range of conclusions from the

integrative approach we have used in this research. In particular, we have definitively determined the atomic resolution structures of two components of the MCIA complex: ACAD9 alone and ACAD9_{WT}-ECSIT_{CTER}. As a result we were able to reveal some large conformational changes that occur during subcomplex formation, determine the importance of the residues involved and therefore identify the exact residues necessary for the deflavination of ACAD9.

We have also definitively determined a mechanism by which we can reverse the ability of ACAD9 to bind ECSIT, and revealed one reason for the specificity of ECSIT to ACAD9, in comparison to structurally similar FAO enzymes.

Furthermore, the experimental evidence of ECSIT phosphorylation is also compelling. Even if ECSIT post-translational modifications were postulated in the seminal article describing the identification of ECSIT as an intermediate in the TL pathway, no specific PTM nor experimental evidence could be provided (Kopp et al. 1999, PMID:10465784). Finally, although the results are exploratory, the effect of amyloid-beta on ECSIT phosphorylation is also unambiguous and provides the first experimental link between ECSIT and amyloid toxicity.

It is also unclear if the findings represent a sufficiently significant advance over their previous paper.

Following on from our previous work where we determined the novel deflavination phenomenon that occurs upon ACAD9-ECSIT interaction, the new atomic level information we provide in this manuscript is invaluable to understanding the structure and function of this key assembly factor. Our experimental approach, involving a wide range of techniques to probe this MCIA subcomplex from its structure in multiple states to *ex cellulo* activity, presents a thorough and complete study of interest to researchers working across a range of fields.

Furthermore, the ACAD9-ECSIT structure that we have determined in this manuscript led to our discovery of an important phosphorylation site, thus revealing a potential mechanism for the regulation of MCIA formation and its role CI assembly linked to mitochondrial dysfunction seen in AD pathogenesis. The presence of ECSIT post-translational modifications and the ensuing effect of amyloidogenic conditions on levels of phosphorylation had until now been completely unknown, and it is essential for this work to be published to inform other researchers, particularly clinical researchers, looking at fundamental molecular causes of neurodegeneration.

In addition, our studies also focus on the ACAD9 homologue VLCAD. Until the work shown in this manuscript the unique ability of ACAD9 to operate in both the FAO and OXPHOS pathways, in comparison to VLCAD which is restricted to the FAO pathway, was not known. Our work has identified the key site that distinguishes the two enzymes, that as commented by reviewer #1 was a remarkable achievement.

I also have a number of questions and comments as listed below.

Comments:

1) If there is an abundance of FAD present, can it eject ECSIT from its binding site on ACAD9?

We thank reviewer #2 for asking this question. We previously investigated whether a significant concentration of FAD in solution could reverse the binding of ACAD9-ECSIT in Giachin et al., using an in vitro Acyl-CoA dehydrogenase assay based on the fluorescence reduction of the electron transfer flavoprotein (ETF) redox partner. Interestingly, we saw that the ECSIT_{CTD} prevented the refluorination of FAD by ACAD9, indicating that once the ACAD9-ECSIT subcomplex is formed, the process is irreversible. As this was already discussed previously, and we refer to Giachin et al. throughout our manuscript, we considered this point irrelevant for inclusion in this manuscript.

2) What would be a proposed mechanism for how excess amyloid beta would cause decreased phosphorylation of ECSIT?

Given the multiple abilities of ECSIT as an integrating hub and a switch regulator, ECSIT could conduct the beneficial or detrimental effects that A β might execute depending on A β levels and subcellular localisation. Although there are studies showing that soluble oligomeric A β is sufficient to dramatically alter MAPK pathways, before amyloid deposition (Palavicini et al 2017, PMID: 28750656), other studies show that the phosphorylation of the Extracellular-signal-regulated kinase (ERK), one representative of the MAPK pathway, is significantly reduced in neuroblastoma cells exposed to β -amyloid, while the phosphorylation of p38 MAP kinase is dose-dependently increased (Daniels et al. 2001, PMID: 11769330). Therefore, the molecular mechanisms leading to the dephosphorylation of ECSIT in a particular amyloidogenic condition can be very diverse, either directly regulated through downstream kinases or by the crosstalk with other signalling pathways, in an attempt to increase the generation of energy, inducing a vicious cycle that could lead to cognitive deficiencies and neurodegeneration.

We have included some of these speculations in the text, although, considering the highly speculative nature of these proposals, we believe it may form the basis for further study, with a broader scope of that of this publication.

3) Does the reduced affinity between phospho-ECSIT and ACAD9 (in figure 6) lead to greater FAD binding in the FAD binding site of ACAD9?

The reduced affinity does not necessarily increase the binding of FAD in the active site of ACAD9, but rather reduces the defluorination ability of ECSIT. As the phosphorylation of Thr320 would prevent the binding of ECSIT to ACAD9, ECSIT is therefore unable to cause the defluorination that occurs when the subcomplex is formed. We believe this is already described in the manuscript discussion and Figure 6. In addition, this is also related to the results that we report where we determine the 15-residue peptide observed in the cryo-EM structure of ACAD9_{WT}-ECSIT_{CTER} is really the key sequence causing the defluorination of ACAD9.

4) Is there evidence for ACAD9 being phosphorylated within the ECSIT binding domain, leading to a similar disruption of the hydrogen bonds between Gln331 and T320 that is observed when the phosphomimetic was made for ECSIT?

In our study we did not focus on the investigation of phosphorylation of ACAD9. This was due to the fact that ECSIT has been identified as a molecular node interacting with enzymes producing A β and therefore links to AD pathogenesis have been proposed. Taking this into account, in MCI complex activity and subsequent CI assembly, ECSIT can be considered a more likely candidate as a regulator of this process. In addition, ECSIT is known as a key participant in the MAPK signaling pathway, where its role is related to several important kinases (Kopp et al. 1999, PMID:10465784). However, the phosphorylation of ACAD9 may form the basis for further study, with a broader scope of that of this publication.

5) It was unclear to me if the choice of kinases used for the study was adequately justified e.g. MAPK p38 alpha. Would one see similar results with a different MAP kinase? Is there any genetic evidence of change in FAO or OXPHOS in cells with p38 MAPK KO (e.g. with CRISPR)?

MAPK p38 alpha is a well-established stress-activated protein kinase, with a role in the inflammation and immune response pathway. It is activated in the response to reactive oxygen species (ROS) and has also been shown to function as a redox sensor, being of an extraordinary importance for the regulation of cellular viability. When and how p38 α is activated, and which targets are engaged, depends on the context (Canovas & Nebreda 2021, PMID: 33504982).

Our rationale for choosing p38 for our *in vitro* phosphorylation studies relies on the work from Kopp and colleagues (Kopp et al. 1999, PMID:10465784), where they showed that ECSIT appeared as a slower migrating band (a non-identified modification) upon cotransfection with MEKK-1 (mitogen-activated protein kinase/ERK kinase kinase-1). Interestingly, MEKK1 propagates downstream signal transduction through the activation of MKK6 (among other kinases), which turns to be an upstream activator of p38 (Witowsky & Johnson, 2003, PMID12456688). Furthermore, p38 α has been implicated in several neuronal functions that are relevant for brain physiology and has also been detected in the early stages of Alzheimer's. In fact, inhibition of p38 α attenuates neuroinflammation, which correlates with improved spatial memory in mouse models of AD (Canovas & Nebreda 2021, PMID: 33504982).

From a more technical point of view, we have a well-established protocol to produce recombinant p38 in both active and non-active forms, in addition to the production of a constitutively active form of MKK6, the p38 activating kinase (described in Juyoux et al. 2023, bioRxiv 2022.07.04.498667), as we show in Figure 6B. Therefore, we could produce kinase samples of top quality that were also ensuring the reliability of the experiments and the results.

For all the reasons mentioned above, we rationalised that p38 could indeed be a good candidate to investigate the potential phosphorylation of ECSIT *in vitro*. Of note, and as observed in Figure 6B, ECSIT phosphorylation is specific of p38, since neither MKK6 nor inactive p38 are able to phosphorylate ECSIT *in vitro*.

We are mindful that these results are preliminary and that the identification of the potential ECSIT kinase will require a much broader portfolio of experiments, which are beyond the scope of this publication.

We have now included the rationale for the selection of p38 as a kinase for our *in vitro* studies in the manuscript.

6) While modeling, a portion of ECSIT (the C terminal residues 221-431) is used to say that there is a conformational change in ACAD9 that ejects FAD in the beta 1 and beta 2 loops adjacent to the FAD binding site. Since this is a portion of ECSIT, is it possible that the conformational change can occur because the rest of the globular ECSIT protein is not present to cause steric hindrance or another phenomenon to prevent this conformational change observed with the C terminal residues?

Due to the positioning of the ECSIT residues (319-335), it is not possible for the β 1- β 2 loop to exist in the 'closed' conformation observed in the ACAD9 structure as there is a clash of atomic positions, namely residues Tyr327 and Glu323 of ECSIT overlap directly with Ser187 and Ala189 of ACAD9, respectively. Therefore we can confirm that even with the full-length ECSIT, the β 1- β 2 loop must be displaced upon subcomplex formation and it is not possible that this can be prevented with full length ECSIT. Additionally, as discussed in previous work published in Giachin et al., no direct interaction occurs between the N-terminal of ECSIT and ACAD9.

Other Comments:

7) What are the loading controls for the phosphorylation assays?

We thank reviewer #2 for bringing up this important point. In the new Figure S6, we have included the Coomassie-stained gels of the four phosphorylation experiments (radiolabelling and in-gel digestion mass spectrometry – *in vitro*, *ex cellulo*-, respectively). Those gels show the loaded samples in each reaction point. Furthermore, for the *ex cellulo* experiments, we applied the LFQ (label-free quantitation) intensities for sample normalisation to best represent the ratio changes of different samples. We have included this information in the description of the mass spectrometry data processing.

8) Why is supplemental figure 6 a supplemental figure rather than a main figure?

We thank reviewer #2 for this suggestion. We have now included Figure S6A into main Figure 6 to complement the other (now Figure 6E).

9) Where are supplemental tables 2 and 3?

Table S2 and S3 correspond to the extra phosphorylation information (excel spreadsheets). These were submitted along with the article.

10) Why doesn't one observe endogenous ECSIT phosphorylation in figure 6F?

That would be explained by the large amount of recombinant ECSIT with respect the amount of endogenous ECSIT that might be present in the cellular extracts, which makes undetectable endogenous phosphorylation.

11) Were any statistics for the phosphorylation measurements done in figure 6?

As mentioned in the comment below, we carried out two independent replicas for each phosphorylation assay, and therefore we could not apply a statistical analysis. However, as described in the methods, each step of the procedure was control monitored to ensure the robustness and the validity of the resulting data.

12) What is the sample size/n for the biological experiments?

As mentioned above, each experiment was done in 2 independent replicas given that we applied several quality metrics to monitor the peak intensities in each experiment and the subsequent data processing to ensure the accuracy of the results, as described in the methods.

Furthermore, the *ex cellulo* assays were very laborious and turned out to be very challenging since the retention of the phosphopeptides in the liquid chromatography column is very close to the end of the elution gradient and therefore, require a more specific procedure than for a conventional LC-MS analysis. We included a manual inspection analysis in the *ex celullo* mass spectrometry data to be consistent with the *in vitro* data analysis, which has strengthen our conclusions. We have updated Supplementary Table S3 with the new analysis and the results are shown in Figure 6G.

Reviewers' Comments:

Reviewer #1:

Remarks to the Author:

The authors adequately addressed technical comments and included movies, which help to understand the interplay between ECSIT and FAD binding.

The points 3 and 4 require further attention.

1) Lines 93-103, where the mutagenesis of ECSIT and ACAD9 is described, now reads as a random set of mutants that are difficult to interpret in terms of the contributions of specific interactions to the binding of the peptide to ACAD9. My suggestion to the authors is to better explain the rationale for choosing the residues to be mutated and summarize what is learned from these mutations at the end of the paragraph. Now it is not clear why the W324A mutant disrupts the complex formation, but not the S191A mutant on the ACAD9 side; why does Y328A mutant form the complex but not the more conservative Y328F mutant?

E323, which I have mentioned previously forms a salt bridge and thus is expected to have a significant contribution to the binding energy. Its weak sidechain EM density does not necessarily suggest a weak interaction but is likely due to radiation damage known to affect particularly strongly carboxyl groups in the proteins.

In the panel G of Figure 2, the positions of peaks for Y327F and Y328A mutants are ambiguous, it should be explained to the readers how these measurements were interpreted and why.

2) The new supplementary movies clarify the positions of the b1-b2 loop, ECSIT peptide, and FAD in 3D. It remains a mystery though that FAD and ECSIT can not co-exist because no obvious clashes are visible between the 2 binders. It is clear how ECSIT peptide changes the conformation of the loop, but neither the loop nor ECSIT seems to clash with FAD. Here a more detailed analysis of FAD-ACAD9 interactions may help to clarify the reason for the FAD dissociation.

3) The last section of experimental results, showing the effect of amyloid on ECSIT phosphorylation is interesting, but in my opinion, is too preliminary to be included in the manuscript. Either these results should be omitted, or further experimental investigation uncovering reasons for the observed effect should be included.

Minor point: In Figure 3F RMSD must be replaced with "Ca-Ca distance". These values are calculated from 2 structures only.

Point-by-point responses to the Reviewer's comments

We appreciate the efforts of the Reviewer in evaluating our work and providing constructive feedback. We are grateful for the opportunity to address the comments and suggestions in order to enhance the quality of our manuscript.

In the light of these we have revised our submission as follows:

Following the reviewer's suggestion, we have produced an extra ECSIT mutant and have incorporated the resulting data into our examination of ACAD9-ECSIT_{CTER} interactions.

Figure 2 has been revised to include the new mutant data.

We have clarified the selection of mutants and have included a supplementary table (**Table S2**) summarising the outcome.

Moreover, we have provided a more comprehensive account of the interactions within the FAD binding pocket in both the open and closed ACAD9 conformations. Additionally, we have conducted a more in-depth analysis of the interactions involving the ACAD9 β 1- β 2 loop with neighbouring residues in each conformation. To facilitate comprehension, we have included an extra figure (new **Figure S5**) illustrating all pertinent interactions within the FAD binding pocket of ACAD9, both in the closed and open conformations, and have included a comparison with VLCAD.

We have corrected **Figure 3F** as the Reviewer highlighted.

We have carefully read the text with the aim to detect and correct typographical or grammatical errors that we might have overlooked in the original manuscript.

For the sake of clarity, all the insertions and modifications, including – where appropriate – more recent references, are highlighted in yellow throughout the manuscript.

Please, find below specific answers to Reviewers' remarks.

Reviewer: The authors adequately addressed technical comments and included movies, which help to understand the interplay between ECSIT and FAD binding.

We thank reviewer for the positive feedback regarding our manuscript. We have taken on board the comments and suggestions and we hope that this clarifies some of the points that were considered unclear before.

The points 3 and 4 require further attention.

1) Lines 93-103, where the mutagenesis of ECSIT and ACAD9 is described, now reads as a random set of mutants that are difficult to interpret in terms of the contributions of specific interactions to the binding of the peptide to ACAD9. My suggestion to the authors is to better explain the rationale for choosing the residues to be mutated and summarize what is learned from these mutations at the end of the paragraph. Now it is not clear why the W324A mutant disrupts the complex formation, but not the S191A mutant on the ACAD9 side; why does Y328A mutant form the complex but not the more conservative Y328F mutant?

E323, which I have mentioned previously forms a salt bridge and thus is expected to have a significant contribution to the binding energy. Its weak sidechain EM density does not necessarily suggest a weak interaction but is likely due to radiation damage known to affect particularly strongly carboxyl groups in the proteins.

In the panel G of Figure 2, the positions of peaks for Y327F and Y328A mutants are ambiguous, it should be explained to the readers how these measurements were interpreted and why.

We thank the reviewer for their suggestion to improve the readability of this section. We have reworded the paragraph in question (page 5), including a summary at the end of the paragraph, and provided an additional summary table (**Table S2**) in order to provide an overview of the rationale for choosing each mutant and the results seen.

In response to this reviewer's point concerning the E323 residue forming a salt bridge with K228 of ACAD9, we designed a E323A mutant in order to investigate the effect of removing this interaction on the stability of the complex. Interestingly, as the reviewer suspected, despite that the Glu323 sidechain seems to be flexible and solvent exposed, the E323A mutant totally abrogates the binding to ACAD9. We have included the mass photometry (MP) and DLS data in the **new Figure 2** and further discuss the relevance of this residue in the ACAD9-ECSIT interface. Figure 1 has also been updated to highlight the E323 residue (**Figure 1 E, F**).

However, taking into account the range of mutants we have studied, we can conclude that it is a concerted effect of multiple residues that influence the formation of the ACAD9-ECSIT_{CTER}, which we demonstrate with our multidisciplinary analysis and discuss in the paper.

We are unsure about the point raised about the positions of the peaks for Y327F and Y328A mutants in Figure 2G. We assume that the reviewer is commenting on the error bars shown for the DLS measurements and not the position of the main data points. However, we believe that the DLS data reflects the results observed in the mass photometry experiments: both the Y327F and Y328A mutants show less perturbation of the complex formation than the other mutants studied. In this case, we have used DLS as a comparative technique in order to evaluate the effect of the mutants on complex formation. However, as demonstrated by the MP data, additional species are also present, which may contribute to the error. We have modified **Figure 2H** to increase the clarity of the different data points.

2) The new supplementary movies clarify the positions of the b1-b2 loop, ECSIT peptide, and FAD in 3D. It remains a mystery though that FAD and ECSIT cannot co-exist because no obvious clashes are visible between the 2 binders. It is clear how ECSIT peptide changes the conformation

of the loop, but neither the loop nor ECSIT seems to clash with FAD. Here a more detailed analysis of FAD-ACAD9 interactions may help to clarify the reason for the FAD dissociation.

We thank the reviewer for complimenting the supplementary movies that we provided after the initial review. We have provided another additional figure (**Figure S5**) that compares the FAD bonding interactions between ACAD9 unbound (i.e. closed conformation), ACAD9 in complex with ECSIT (i.e. open conformation) and VLCAD with additional analysis in the text. As shown in the new figure, there are numerous interactions between the FAD and VLCAD, leading to a more stable FAD moiety than in ACAD9. We have also labelled the β 1- β 2 and α 11'- α 12 loops in **Figure S5C** to highlight the β 1- β 2 loop stabilising interface in the overall ACAD9 homodimeric structure. Furthermore, an overlay of a FAD molecule with the ACAD9-ECSIT model reveals that FAD in that conformation has even less contacts with ACAD9 than in the closed conformation. Interestingly, ECSIT Glu323 replaces one of the hydrogen bonds between the β 1- β 2 loop and the β 4 strand that helps to stabilise the loop in the down/closed conformation (**Figure S5 D-E**). Therefore, we speculate that ECSIT might actually play a role of an allosteric modulator by stabilising, in this way, the open conformation with a lower FAD binding affinity. That would explain why FAD and ECSIT cannot co-exist despite no obvious clashes. We have included these thoughts in the Discussion, page 12.

3) The last section of experimental results, showing the effect of amyloid on ECSIT phosphorylation is interesting, but in my opinion, is too preliminary to be included in the manuscript. Either these results should be omitted, or further experimental investigation uncovering reasons for the observed effect should be included.

We acknowledge the specific point raised by the reviewer about the experimental results related to the effect of amyloid on ECSIT phosphorylation, and we understand the reviewer's concern about the preliminary nature of these findings.

However, we would like to emphasize the importance of these results in the context of our study. While we agree with the reviewer that further experimental investigations are necessary to fully elucidate the underlying mechanisms of the observed effect, we believe that sharing this information with the wider scientific community is valuable. The effect of amyloids on ECSIT phosphorylation opens up new avenues for research, including clinical studies, and could have implications for understanding the pathway coordination in mitochondrial complex I assembly.

Therefore, we think that these results should be included in the manuscript. We understand the need for thorough and comprehensive research but we believe that by including these results, we can stimulate interest and collaboration within the scientific community to explore this direction further. We have included a clear statement acknowledging that these results are exploratory and warrant additional experimentation to determine how exactly amyloids contribute to the ECSIT and ACAD9 functional activities (page 13).

4) Minor point: In Figure 3F RMSD must be replaced with "Ca-Ca distance". These values are calculated from 2 structures only.

We again thank reviewer #1 for noticing this slight oversight. We have replace the RMSD by Ca-Ca distance in **Figure 3F**.